

# Exploring the regolith with electrical resistivity tomography in large-scale surveys: electrode spacing related issues and possibility

Laurent Gourdol[1], Rémi Clément[2], Jérôme Juilleret[1], Laurent Pfister[1], Christophe Hissler[1]

[1] Catchment and Eco-hydrology Research Group (CAT), Luxembourg Institute of Science and Technology (LIST), Belvaux, L-4422, Luxembourg
[2] REVERSAAL Research Unit, National Research Institute for Agriculture, Food and Environment (INRAE), Villeurbanne, F-69626, France

*Correspondence to*: Laurent Gourdol (laurent.gourdol@list.lu)

**Abstract.** Within the Critical Zone, regolith plays a key role in the fundamental hydrological functions of water collection, storage, mixing and release. Electrical Resistivity Tomography (ERT) is recognized as a remarkable tool for characterizing
the geometry and properties of the regolith, overcoming limitations inherent to conventional borehole-based investigations. For exploring shallow layers, a small electrode spacing (ES) will provide a denser set of apparent resistivity measurements of the subsurface. As this option is cumbersome and time-consuming, smaller ES – albeit offering poorer shallow apparent resistivity data – are often preferred for large horizontal ERT surveys. To investigate the negative trade-off between larger ES and reduced accuracy of the inverted ERT images for shallow layers, we use a set of synthetic "conductive / resistive /
conductive" three-layered soil–saprock/saprolite–bedrock models in combination with a reference field dataset. Our results suggest that an increase in ES causes a deterioration of the accuracy of the inverted ERT images in terms of both resistivity distribution and interface delineation and, most importantly, that this degradation increases sharply when the ES exceeds the thickness of the top subsurface layer. This finding, which is obvious for the characterization of shallow layers, is also relevant even when solely aiming for the characterization of deeper layers. We show that an oversized ES leads to
overestimations of depth to bedrock and that this overestimation is even more important for subsurface structures with high resistivity contrast. To overcome this limitation, we propose adding interpolated levels of surficial apparent resistivity relying on a limited number of ERT profiles with a smaller ES. We demonstrate that our protocol significantly improves the accuracy of ERT profiles when using large ES, provided that the top layer has a rather constant thickness and resistivity. For the specific case of large-scale ERT surveys the proposed upgrading procedure is cost-effective in comparison to protocols
based on small ES.



## 1 Introduction

Within the Critical Zone, the architecture and properties of the regolith, as well as its distribution across the landscape, play a key role in how rainfall is collected, stored and finally released to generate streamflow (Schoeneberger and Wysocki, 2005; Lin, 2010; Ghasemizade and Schirmer, 2013; Brooks et al., 2015). Factors such as the depth and composition of the soil

cover and the rock weathering determine water pathways, storage capacity, residence times in the subsurface and subsequent interactions with surface water bodies (Freer et al., 2002; Hopp and McDonnell, 2009; Graham et al., 2010; Gabrielli et al., 2012; Lanni et al., 2013; Ameli et al., 2016).

However, limited access to the subsurface is a major hurdle to acquiring this information meaning that often, even the most basic data is missing, such as the transitions from the soil to the hard bedrock (Brooks et al., 2015). It is the complexity and

spatial variability of the subsurface that make its characterization very challenging. Conventional investigation techniques (i.e. soil pits, drillings) of regolith are known to be invasive and of limited spatial representativeness – a trait causing them to be ignored in the vast majority of catchment studies (Burt and McDonnell, 2015; Parsekian et al., 2015). Several authors have also recently pointed out the subsurface as being the greatest knowledge gap in the understanding/modelling of hydrological processes, with a greater investment into "seeing" the subsurface needed to provide the Earth System

Modelling community with critical guidance on how to parameterize model subsurface structure depths and properties (Fan et al., 2019).

Geophysical prospection techniques have received increasing attention in recent years within the hydrological sciences community, thanks to their non-destructive character and ability to provide information on subsurface features over large areas. These investigative tools are now recognized as being essential for accurately characterizing the subsurface and

studying water partitioning (Robinson et al., 2008; Loke et al., 2013; Binley et al., 2015; Brooks et al., 2015; Parsekian et al., 2015; Singha, 2017). Among the geophysical prospection techniques at hand, electrical resistivity tomography (ERT) is commonly used to characterize subsurface environments. This well-known technique is based on the injection of an electrical current through a pair of electrodes and the measurement of the resulting electrical potential between a second pair of electrodes along a line of dozens or hundreds of grounded electrodes. Through inversion schemes, ERT data is used to

generate 2D and 3D electrical resistivity maps of the subsurface (see e.g. Binley and Kemna, 2005 for a detailed explanation of the ERT method).

The electrical resistivity of the subsurface provides a weighted average of the electrical properties of its mineral grains, liquid and air (Archie, 1942; Keller and Frischknecht, 1966; Reynolds, 2011). Constitutive relationships can be used to link electrical resistivity to several properties and states that are of major interest to hydrologists: e.g. textural properties (Tetegan

et al., 2012), porosity (Leslie and Heinse, 2013; Comte et al., 2018), hydraulic conductivity (Slater, 2007; Farzamian et al., 2015), water content (Brunet et al., 2010; Alamry et al., 2017) or solute concentrations (Bauer et al., 2006; Comte and Banton, 2007). While these constitutive relationships are essential for reliable hydrological interpretations (Binley et al., 2015), their accuracy largely depends on the resolution of the ERT images (Day-Lewis et al., 2005).



ERT has also been successfully used to characterize regolith architecture by delineating areas showing similar resistivity patterns (Crook et al., 2008; Comte et al., 2012 ; Leopold et al., 2013; Cassidy et al., 2014; Holbrook et al., 2014; Hübner et al., 2015; Uhlemann et al., 2015; Wainwright et al., 2016; Scaini et al., 2017). An increasing number of studies use automated edge detection approaches to delineate these key interfaces within the subsurface (Nguyen, 2005; Hsu et al., 2010;

Chambers et al., 2012, 2013, 2014, 2015; Audebert et al., 2014; Ward et al., 2014; Uhlemann et al., 2015; Wainwright et al., 2016; Scaini et al., 2017). However, it has also shown that the application of these methods can fail – even when the true interface is sharp – because of insufficient sensitivity and accuracy in the vicinity of the interface (Chambers et al., 2013, 2014).

Ultimately, the geophysical information on the subsurface that can be derived from ERT investigations, either in terms of

geometry or hydraulic properties, may be used to feed process-based hydrological/hydrogeological models in order to improve their rightness/realism in terms of spatial variability (e.g. Mastrocicco et al., 2010; House et al., 2016; Loritz et al., 2017; Comte et al., 2018). However, when resistivity models form the basis for constructing such process-based models, the accuracy of the geophysical information and its interpretation is a critical issue that can lead to seriously wrong models and correspondingly wrong model predictions (Andersen et al., 2013).

The characterization of subsurface properties and the delineation of structural units within it should thus go hand in hand with a suitable resolution of ERT images. Otherwise, the results can be inaccurate (Chambers et al., 2013, 2014; Clément et al., 2009, 2014; Ward et al., 2014). Chambers et al. (2014) emphasize that using ERT to detect thin surficial layers remains challenging. Indeed, when shallow structures are investigated, a small electrode spacing (ES) is required, as it delivers denser and well-discretized measurements of the shallow subsurface (Reynolds, 2011; Chambers et al., 2014). However, for

large horizontal ERT surveys, small ESs are often not a viable option, as their implementation remains time-consuming. Depending on the regolith architecture and the size of the investigated area, it is thus challenging if not impossible to balance the requirement for shallow layers characterization (i.e. smaller ESI), against the competing need to cover the area of interest within a reasonable amount of time and cost (i.e. larger ESI) (Chambers et al., 2014). Moreover, as shown by Kunetz (1966) or Clément et al. (2009), oversizing the ES might not only be inappropriate for the characterization of thin surface layers, but

it may also affect the characterization of deeper layers by causing a depth-based resistivity bias, as a result of the inversion process which can be affected by a lack of shallow data.

In this study, we focus on which ES should be considered to characterize the entire regolith accurately. More specifically, we are concerned whether deep structures are well defined (within the limits of the intrinsic resolution limitation of the ERT method that translates in a drop in resistivity model sensitivity with depth) if the shallow structure is not well sampled (i.e. if

a larger ES is used). These issues should be ideally addressed prior to the design of fieldwork campaigns to avoid any misinterpretation of field data. Ultimately, we aim at carefully chosing the ES parameter for adequately disentangling the subsurface architecture and properties (both for the shallow part of the subsurface and for the deeper layers). We intend to increase the potential for large horizontal ERT surveys – based on oversized ES – to equally deliver detailed knowledge on shallow subsurface structures. Within this work, we investigate:



1) How the inverse solution reconstruction is affected by the ES parameter (i.e. impact of the lack of shallow apparent resistivity data induced by the use of an oversized ES on inverted ERT image accuracy) and which is the most appropriate ES value for accurately characterizing the entire regolith (i.e. for both surface and deeper horizons);

2) The potential for a new approach to improve the accuracy of ERT surveys relying on a large ES by adding interpolated

levels of surficial apparent resistivity based on a limited number of measurements with a small ES.

To this end, we use as a reference case study the Weierbach catchment, where an earlier ERT survey has been carried out in order to shed new light on the spatial heterogeneity of the subsurface, but for which we have been facing ES choice related issues. In addition to the field dataset, we investigate a set of synthetic soil–saprock/saprolite–bedrock models using a classical geophysical approach based on numerical modelling to corroborate and reinforce the results. While the former

represents a field reality in terms of heterogeneity, the latter provides important information under controlled conditions and *a priori* exact knowledge. The assessment of the ERT images obtained from these two datasets is carried out considering the accuracy of the inverted resistivity distribution and the derived interface depths.

## 2 Materials and Methods

### 2.1 Field study

#### 2.1.1 Study area description

Our experimental test site is the forested Weierbach headwater catchment, located in the Luxembourgish Ardennes Massif (0.45 km²; Figure 1). The geological substratum of the study area is composed of Devonian metamorphic slates. Recent studies in the Weierbach catchment have shown that its regolith plays a key role in runoff generation processes (Pfister et al., 2010; Wrede et al., 2015; Martínez-Carreras et al., 2015; Martínez-Carreras et al., 2016; Scaini et al., 2017, 2018), including

lumped-parameter and process-based modelling studies (Fenicia et al., 2014; Glaser et al., 2016, 2019, 2020). Hence, its characterization is of high relevance for gaining new insights into the fundamental catchment functions of water collection, storage and release (Pfister et al., 2017). Several soil pits and drillings were done in the catchment in order to describe its regolith structure and mineralogical and chemical properties (Figure 1). Based on the visual inspection of soil pits and core drillings and particle size distribution analysis and porosity measurements (Juilleret et al., 2011; Wrede et al., 2015; Juilleret

et al., 2016; Martínez-Carreras et al., 2016; Moragues-Quiroga et al., 2017; Scaini et al., 2017), Figure 2a-c shows a mean schematic representation of the soil-to-substratum continuum. According to rock weathering and pedological processes (Velde and Meunier, 2008; Juilleret et al., 2016), this structure can be partitioned into three main units characterized as follows:

1) The solum is a stony loam soil with a mean thickness of 50 cm. The loam texture stems from a loess deposit, which was

mixed through solifluction with many slate clasts native from the bedrock (coarse element content around 25%). The solum has a high drainage porosity of 30% on average.





2) The subsolum has two parts. The upper subsolum (from 50 to 90 cm depth on average) is a loam to sandy-loam texture layer with abundant slate fragments. In this part, the abundance and size of rock fragments strongly increases with increasing depth (coarse element content increases from 30% to 75%). Inversely, the drainage porosity decreases, from 30% to 10%. The lower subsolum (from 90 to 140 cm depth on average), with the largest content of slate fragments (coarse element content greater than 80%), corresponds to the decomposed/broken part of the bedrock.

3) The slate hard bedrock starts, on average, at a depth of 140 cm. At first, very large fractures in the hard bedrock tend to close quickly as the depth increases. At a depth of about five metres, most fractures are closed and the bedrock can be considered fresh and almost impermeable.

Given this intrusive point-scale investigation knowledge, the solum/subsolum electrical resistivity interface is expected to be sharp while the subsolum/hard bedrock interface is most probably more gradational. In addition, point-scale observations suggest a probable spatial variability of the subsurface electrical resistivity. Indeed, mirroring multiple weathering phases in the Luxembourgish Ardennes Massif over geological time scales (Moragues-Quiroga et al., 2017; Demoulin et al., 2018), cores obtained from deep drilling campaigns (Figure 2-d) reveal different weathering degrees in the Weierbach catchment (Figure 2-e). The top of the substratum presents a high weathering degree in the upper part of the basin (north and west of the catchment, morphologically expressed by a plateau). Elsewhere in the catchment, bedrock weathering is less pronounced (on hillslope position and along the eastern limit). This difference also implies contrasted surface layer properties. As observed in soil pits on the plateau, slate fragments are smaller and less consistent and the clay content of the matrix is higher (Figure 2-f; Regolithic Saprolite Subsolum type as per Juilleret et al., 2016). Elsewhere, soil pits exhibit bigger and more coherent slate fragments and less clay in the matrix (Figure 2-g; Regolithic Saprock Subsolum type as per Juilleret et al., 2016).

### 2.1.2 ERT survey design, data collection and processing

For a characterization of the subsurface of the entire Weierbach catchment, we built a mesh of several large ERT profiles using the roll-along technique (white lines drawn in Figure 1, cumulative length of about 12 km). The goal was here to inform on the spatial organisation and connectivity of the catchment subsurface in terms of the regolith's weathering state and depth to bedrock and to provide eventually new insights on the substratum further deep. To complete this catchment-wide survey in a reasonable time and meet the targeted objectives (in terms of both horizontal and vertical discretization and depth of investigation), we chose a set-up with an ES of 2 m. In order to characterize more accurately the soil-to-substratum continuum for specific landscape units, we added 12 plot scale ERT profiles of 120 electrodes each, using a smaller ES of 0.5 m (red lines drawn in Figure 1). Their locations were chosen according to prevailing local geomorphological characteristics (plateau, steep and gentle hillslope, interfluve, close to riparian zone). These last 12 profiles are the ones that we rely on in this study to address the research objectives related to the lack of shallow apparent resistivity data induced by the use of an oversized ES (with the goal of improving *in fine* the accuracy of ERT images from the catchment scale survey dataset).



All measurements were taken with a Syscal Pro 120 (ten-channel) resistivity meter from IRIS Instruments with multicore cables attached to stainless steel rod electrodes. A pulse duration of 500 ms and a target of 50 mV for potential readings were set as criteria for the current injection. To insure a good repeatability, stacks numbers were automatically adjusted between 3 and 6 aiming for a maximum standard deviation of 3% for the repeated measurements. We retained the Wenner-

Schlumberger array for the measurements. This option offers good depth determination and spatial resolution (Dahlin and Zhou, 2004). Despite the fact that the Wenner-Schlumberger reciprocal configuration tends to pick up more noise than the normal configuration (Dahlin and Zhou, 2004), we decided to use it because it offers a quick data acquisition time when using a multi-channel resistivity meter. The measurement sequence contains quadrupoles with internal electrodes separations of 1 to 9 times the ES and internal-external electrodes distances of 1 to 8 times the internal electrode separations.

To assess data accuracy, we measured however 25% of the quadrupoles in a normal configuration. Reciprocal errors calculation (defined as the percentage standard error in the average of the normal and reciprocal measurements; Wilkinson et al., 2012), together with the analysis of the standard deviations obtained for the repeated measurements, allow to characterize the measurements as both very precise and accurate (99.4% of the standard deviations do not exceed 1% and 99.1% of the reciprocal errors are below 5%; mean standard deviation and reciprocal error values of 0.10 and 0.68 %, respectively). Even

though the overall quality of the data was good, we applied a cleaning procedure (removal of obvious apparent resistivity outliers and all quadrupoles presenting a measured potential lower than 10 mV or a standard deviation of the repeated measurement higher than 3%). After raw data processing, more than 99.5% of the original dataset remained available for each of the 12 profiles.

First, all available processed apparent resistivity data were used for the inversion of each profile. Second, to match the set-up

of the catchment scale survey and document the associated loss in resolution, only quadrupoles measured with an ES of 2 m (or equivalent quadrupoles in terms of external electrodes distance) were considered.

## 2.2 Synthetic resistivity dataset

### 2.2.1 Conceptual resistivity models

As mentioned previously (see 2.1.1), for a given geological substratum, and according to rock weathering and pedological

processes, the regolith can be partitioned into three main units (Velde and Meunier, 2008; Juilleret et al., 2016), namely, from top to bottom:

1) the solum, which is the "true soil", where pedogenetic processes are dominant,

2) the subsolum, corresponding to weathered materials where the original rock structure is preserved and geogenic processes still dominate (depending on the degree of weathering the saprock and/or saprolite can be distinguished), and

3) the hard bedrock.

Based on this three-layered subsurface conceptual model, and according to the specificity of the Weierbach catchment, we generated 25 one-dimensional "conductive solum / resistive subsolum / conductive bedrock" conceptual models to



investigate different scenarios with varying resistivity and thickness contrasts. Solum and bedrock resistivity was set to 1000 ohm.m for all models. The solum thickness was also set to a unique value of 0.5 m, which is also in line with the average thickness observed in our study area (see 2.1.1). To cover a sufficiently wide range of subsurface structures and properties, the subsolum layer was parameterized with several values of thickness (0.5, 1, 2, 4, 8 m) and resistivity (1250, 2500, 5000,

10000, 20000 ohm.m). The retained resistivity values were also chosen according to the range observed during the field study.

It is worth noting that we have opted for the use of a 1D synthetic model structure, but that the subsequent forward modelling and inversion processes will be then done in 2D in order to evaluate not only the accuracy but also the precision of the inversion results. This would not have been possible using a 1D inversion scheme.

**2.2.2 Forward Modelling, ERT arrays and electrode spacing**

To mimic apparent resistivity measurements with the synthetic models, we simulated the electric field distribution resulting from current injections using the electric field distribution theory (Maxwell's equation) and the finite element method. We performed numerical simulations using the AC/DC module of Comsol Multiphysics, complemented with a forward 3D modelling (F3DM) Matlab script (Clément et al., 2011; Audebert et al., 2014). This script assesses automatically, for a

quadrupole sequence, the electric potential between the two potential electrodes, for a given current. To achieve a realistic dataset reflecting the properties of a field survey, we applied a systematic Gaussian noise distribution with 3% standard deviation relative error to the apparent resistivity dataset to simulate the noise commonly recorded with the resistivity meter. In addition to the Wenner-Schlumberger array which was used for the Weierbach catchment survey, the dipole-dipole array was also use here to simulate apparent resistivity from the resistivity models in order to broaden the modelling findings. The

dipole-dipole and the Wenner-Schlumberger arrays represent the two most commonly used ERT arrays (Carrière et al., 2017). Their successful application in field studies is mainly due to their surveying efficiency and sensitivity (Dahlin and Zhou, 2004). In order to assess the effect of the lack of shallow apparent resistivity measurements related to the ES choice, simulations of apparent resistivity for both arrays were conducted using 5 different ESs (0.25, 0.5, 1, 2 and 4 m).

**2.3 Upgrading apparent resistivity datasets measured with a large electrode spacing**

As illustrated in Figure 3, the use of a larger ES leads to less apparent resistivity measurements that possibly induce a critical lack of shallow apparent resistivity data. In the event of an ERT survey carried out with a large ES – and for which the first acquisition level (i.e. quadrupoles whose external electrodes separation is of the smallest possible extension, see Figure 3) is too deep to properly characterize the subsurface structure's top layer (in our case the solum), we propose to take advantage of the potential relationships between this first acquisition level and additional surficial apparent resistivity acquisition levels

(i.e. quadrupoles with smaller external electrodes separations, see Figure 3) obtained from a reduced number of ERT profiles with a smaller ES. If the top layer has a rather constant thickness and resistivity, we consider that such relationships exist and



could then be transposed to areas where the larger ES have been used and where data gaps prevail in the shallow subsurface. This approach may eventually reduce the oversized ES related effects.

Within this work, we assess the proposed approach by applying it to the ERT profiles relying on an ES of 2 m for both synthetic and field datasets. The protocol for eventually obtaining upgraded ERT datasets is as follows:

1) From the set of apparent resistivity data measured with an ES of 2 m, we extract the first acquisition level of apparent resistivity data (for the smallest possible external electrodes separation, i.e. 6 m). For this acquisition level, we extract – from the set of apparent resistivity data relying on an ES of 0.5 m – four subsets of apparent resistivity data for smaller external electrodes separations of 1.5, 2.5, 3.5 and 4.5 m, respectively.

2) We use these subsets to assess (using regression analysis) if robust relationships exist between the apparent resistivity data

for external electrodes separations of 1.5, 2.5, 3.5 and 4.5 m respectively, and those of the first acquisition level measured with an ES of 2 m.

3) If such relationships exist, we ultimately use the four resulting equations to upgrade each initial ERT profile, relying on an ES of 2 m with four levels of surficial apparent resistivity, which are interpolated from the first acquisition level of apparent resistivity data.

## 15  2.4 Inversion procedure

Both (synthetic and field) datasets were inverted with the same procedure. Inverse solution reconstruction of the interpreted resistivity distribution relied on the BERT code (Boundless Electrical Resistivity Tomography; Günther and Rücker, 2016). This code is based on the finite element-forward modelling and inversion method described in Rücker et al. (2006) and Günther et al. (2006). The aim of the inversion process is to calculate a resistivity model that satisfies the observed apparent

resistivity data. A homogeneous starting model is generated with the median measured apparent resistivity, for which a response is calculated and compared to the observed data. The starting model is then modified iteratively until an acceptable convergence between the model response and the observed apparent resistivity is achieved. The root mean square misfit error (Loke and Barker, 1996) and the $\chi^2$ criteria (Günther et al., 2006) are used to assess the adequacy between the model response and the observed apparent resistivity. While the root mean square misfit error is the normalized root mean square of

the data fit and should be in the range of the relative data error, $\chi^2$ is a measure on how good a model fits the observed data for a given data error and thus this measure scales with the error.

The constraints placed on the resistivity model during the inversion had to be carefully considered. Here, we used the same 2D inversion settings for all apparent resistivity datasets: a smooth inversion optimization method (L2-norm), a z-weight factor of 1 for generating isotropic resistivity distribution and a regularization parameter λ of 20. In many circumstances, an

L1 model constraint with a lower λ value (Loke et al., 2003) would have been preferred for investigating lithological boundaries, since it favours sharp changes in resistivity. But in our case, although the solum/subsolum boundary was expected to be relatively sharp in the Weierbach catchment, the subsolum/hard bedrock interface had a more gradational character. Moreover, the closing with depth of the fractures in the hard bedrock implies also potential smooth changes in





resistivity. For those reasons, an L2 model constraint with a moderate λ value was therefore considered to provide a good compromise. Finally, it is important to note that particular care has been taken in discretizing the models. Indeed, following the standard automatic meshing in the inversion code, the larger the ESI, the coarser the mesh would have been (Günther and Rücker, 2016). As shallow resolution is the main point of our study, and because inversion results are to a certain degree

mesh-dependent, the same fine mesh (whose resolution suits the smallest ES according to Günther and Rücker, 2016) was used for all inversions in order to avoid any coarse-ness meshing issues in the comparison between the resulting interpreted resistivity images.

**2.5 Efficiency criteria for models quality assessment**

For the synthetic dataset, we evaluate the agreement between true synthetic resistivity models and interpreted resistivity

distributions using the Nash–Sutcliffe model efficiency coefficient (NSE):

$$NSE = 1 - \frac{\sum_{i=1}^{n}(O_i - P_i)^2}{\sum_{i=1}^{n}(O_i - \bar{O})^2} \qquad (1)$$

with O for actual data (true synthetic resistivity values), $\bar{O}$ for mean of actual data, P for calculated data (calculated resistivity from inversion process) and n for number of data (number of meshes). Originally developed (Nash and Sutcliffe,

1970) and widely used (Bennet et al., 2013; Hauduc, 2015; Gupta et al., 2009) for hydrological purposes, the NSE coefficient has also been applied to evaluate the quality of several environmental models, such as geophysical models (Tran et al., 2016).

We compared the true interface depths with those that can be derived from inverted ERT images as an additional way to assess the accuracy of the results. In ERT image analysis, isosurface and derivative methods are the two groups of methods

commonly used for this purpose. Here, we retained the group of derivative methods despite it being shown that these methods can fail because of insufficient sensitivity and accuracy in the vicinity of the interface (Chambers et al., 2013, 2014). Indeed, derivative methods represent the most universal way to extract interfaces because their use is relevant both in homogeneous and heterogeneous subsurface contexts (Chambers et al., 2014). It is worth noting also that derivative methods have already been used with success in other ERT studies, even when using an L2-norm (smooth) model constraint (e.g. Hsu

et al., 2010; Chambers et al., 2012; Ward et al., 2014). Derivative methods assume that interfaces are located where changes in image properties are at a maximum. These changes can be detected using either the first or the second derivatives, targeting maximum gradients or zero values, respectively (Marr and Hildreth, 1980; Torreão and Amaral, 2006; Sponton and Cardelino, 2015). In this study, we used the second derivative of ERT images and targeted zero values (e.g. Hsu et al., 2010) with Paraview software (Ahrens et al., 2005). To be consistent with the inverse solutions delivered by BERT (Günther et al.,

2006), we calculated the second derivative on the logarithm of resistivity. Finally, we defined interfaces by following second derivative zero contour continuity and horizontality, as well as the resistivity distribution and its associated gradient (first





derivative). Note that the delineation of some interfaces results from the merging of several zero contours to ensure their continuity.

The same accuracy criteria (NSE and interface depths derived from second derivative zero values) were used for the field dataset of the Weierbach catchment to assess and compare the accuracy of the inverted ERT profiles obtained from the

quadrupoles measured with an ES of 2 m (or equivalents in terms of external electrodes distance), when upgraded or not with the four surficial interpolated levels. In this case, inverted ERT profiles, resulting from the full apparent resistivity measurements using an ES of 0.5 m, served as reference models.

## 3 Results

### 3.1 Synthetic modelling results

The 300 resistivity models resulting from the inversion of the synthetic resistivity models are provided as supporting information (Figures S1-S12). Depending on the models, the inversion process was terminated after 1 to 11 iterations. As indicated by the root mean square misfit error (average: 0.89%, range: 0.40-2.12%) and the $\chi^2$ criteria (average: 0.81, range: 0.16-4.11), acceptable convergence between the calculated and simulated apparent resistivity data was achieved for all models. In 98% of all cases, the root mean square misfit error and the $\chi^2$ were less than 1.5 and 2, respectively.

Note that since the results obtained for both arrays are very similar, we only present those obtained from the Wenner-Schlumberger array and that fit the field case study (Figures 4-6, Tables 1-2). The same figures and tables resulting from analyses carried out on inversion results from the dipole-dipole array have nonetheless been also produced, but are provided in the supplementary material (Figures S13-S15, Tables S1-S2).

### 3.1.1 Impact of the electrode spacing on models accuracy

The visual examination of the inversion results (Figures S1 and S6) and NSE values (Tables 1 and S1, Figures 4 and S13) obtained for the smallest ES (0.25 m) indicate an overall good match of the ERT images with synthetic resistivity models serving as benchmarks. Mean NSE values for the 25 synthetic models are equal to 0.60 and 0.61, respectively for the dipole-dipole and the Wenner-Schlumberger arrays. As indicated by the mean NSE value of 0.55 for both the dipole-dipole and the Wenner-Schlumberger arrays, results for an ES of 0.5 m are slightly less positive (Table 1 and S1, Figures 4 and S13).

Nonetheless, here again the resistivity distributions obtained from inversions show a good reproduction of the synthetic resistivity models (Figures S2 and S7). Regarding interface delineation, the results obtained from ERT images with ESs of 0.25 and 0.5 m are also good (with a slightly better accuracy when using the smallest spacing) and offer a good reproduction of the solum thickness and depth to bedrock, as revealed by the proximity of estimates with true depths (Tables 1 and S1, Figures 4 and S13). Considering the 25 synthetic models overall, for ESs of 0.25 and 0.5 m, the mean differences observed

for the solum depth are 0.05 and 0.06 m using the dipole-dipole array and 0.02 and 0.04 m using the Wenner-Schlumberger array, respectively. For depth to bedrock, for ESs of 0.25 and 0.5 m, observed mean differences reach 0.05 and 0.19 m using





the dipole-dipole array and 0.27 and 0.34 m using the Wenner-Schlumberger array, respectively. We note from the results of the two smallest ESs that resistivity and thickness contrasts of the synthetic resistivity models influence the accuracy of inverted models. Indeed, when the resistivity contrast of the subsolum is too low or too high (i.e. 1250 and 20000 ohm.m), the NSE values are lower (Tables 1 and S1, Figures 4 and S13). Similarly, the NSE values also indicate slightly worse results

when the subsolum is thin (i.e. 0.5 m). Resistivity contrasts also affect the delineation of interfaces. We observed that an increase in the resistivity contrast induces an overestimation effect of the interface depths (Tables 1 and S1, Figures 4 and S13). This last finding is less obvious at deeper depths to bedrock.

Although the information delivered when using an ES of 1 m is still valid to estimate the synthetic resistivity models, its accuracy is significantly weakened in comparison to that obtained with ESs of 0.25 and 0.5 m (i.e. mean NSE values for the

25 synthetic models of 0.33 and 0.34 for the dipole-dipole and the Wenner-Schlumberger arrays, respectively; Tables 1 and S1, Figures 4 and S13). The visual examination of the inversion results indicates an increase of local artefacts induced by the resolution degradation (for instance, see inversion results in Figures S3 and S8 when the subsolum resistivity and thickness in the synthetic model are equal to 1250 ohm.m and 8 m, respectively). This degradation is mainly restricted to the lowest resistivity contrast and therefore does not explain the general decrease in the accuracy of the results. For the strongest

resistivity contrasts, the inversion process leads to relatively well-defined three-layered structures. However, these are shifted down in depth, in comparison to the synthetic resistivity models (especially for the solum-subsolum interfaces; Tables 1 and S1, Figures 4 and S13). For the 25 synthetic models overall, we observed a mean overestimation of 0.30 and 0.33 m for the solum depth using the dipole-dipole and the Wenner-Schlumberger arrays, respectively. Similarly, the depths to bedrock are overestimated by an average of 0.32 and 0.54 m for both arrays, respectively. When looking at the subsolum

characteristics in detail, the deepening effect on the obtained structure is more pronounced as the resistivity of the subsolum is higher and thicker (Tables 1 and S1, Figures 4 and S13).

Finally, mean NSE values for the 25 synthetic models obtained from ERT images using ESs of 2 and 4 m are close to (-0.03 and 0.00 for the dipole-dipole and the Wenner-Schlumberger arrays, respectively) and less than zero (-0.20 and -0.12 for the dipole-dipole and the Wenner-Schlumberger arrays, respectively). This indicates an overall performance that has not

improved, in the first case, and is even worse, in the second case, than when simply using the mean of the synthetic resistivity models. As shown by the inversion results (Figures S4-S5 and S9-S10), several artefacts disturb the quality of ERT images, predominantly (but not exclusively) when the resistivity contrast is low. We also observed that the distinction between solum and subsolum is not obvious, not only when the subsolum is thin, but even more so when the contrast in resistivity is low. In these cases, the NSE value is always lower than zero and, due to the badly resolved structures, interface

delineation from the second derivative of the ERT images often result from merging several second derivative zero contours (Tables 1 and S1, Figures 4 and S13). The analysis of the derived interface depths clearly shows that the precision of the interfaces is worse (especially for an ES of 4 m as indicated by the large standard deviations) and, even more important, their accuracy with respect to true depths is poor (Tables 1 and S1, Figures 4 and S13). Taking into consideration all resistivity and thickness contrasts and using ESs of 2 and 4 m, the mean differences observed for the solum indicate an overall





overestimation of 0.60 and 0.81 m using the dipole-dipole array, and 0.63 and 0.75 m when using the Wenner-Schlumberger array, respectively. Looking at subsolum characteristic differences, an overestimation of the solum depth is greater as the resistivity of the subsolum is high and it is thicker. Furthermore, for low resistivity and thin subsolum, the delimited interfaces of the solum were often characterized by zero depth (Tables 1 and S1, Figures 4 and S13). Hence, we note a

skewing of the mean difference toward negative values. Concerning the depth to bedrock, its estimation is also strongly dependent on the subsolum characteristics of the synthetic resistivity models, leading to a weak and spread correlation with true depths (Tables 1 and S1, Figures 4 and S13). In most cases, we observed an overestimation. The overestimation increases as the contrast in resistivity in the model becomes larger and the true depth to bedrock gets lower. Conversely, lower resistivity contrasts and deeper true depths to bedrock lead to larger underestimated values.

**3.1.2 Application and assessment of the proposed approach to upgrade ERT datasets**

As shown in the scatter plots of Figures 5 and S14, each of the four selected surficial apparent resistivity levels acquired with an ES of 0.5 m (vertical axes) can be derived from the first apparent resistivity acquisition level using an ES of 2 m (horizontal axes) assuming a linear interpolation, whether for the Wenner-Schlumberger array or the dipole-dipole array. As indicated by low root mean square relative error values, the accuracy of each linear regression is good, regardless of the

array and the surficial acquisition levels. From the equations of these linear regressions, the resulting four interpolated levels of surficial apparent resistivity were added to the apparent resistivity datasets using the ES of 2 m. ERT images resulting from the inversion of these upgraded datasets are provided in Figures S11 and S12 in the supplementary material for the Wenner-Schlumberger and the dipole-dipole arrays, respectively. The accuracy criteria, allowing the assessment of their efficiency to reproduce true synthetic models, are shown in Tables 2 and S2 and Figures 6 and S15. Here again, results are

fairly comparable between the dipole-dipole and the Wenner-Schlumberger arrays.

The visual examination of the inversion results (Figures S11-S12) and NSE values obtained using the four surficial interpolated levels indicate an overall good match between the ERT images and synthetic resistivity models (Table 3, Figure 7). Mean NSE values for the 25 synthetic models for the dipole-dipole and the Wenner-Schlumberger arrays are equal to 0.34 and 0.35, respectively. These values are much better than those obtained when using the standard apparent resistivity

datasets (i.e. -0.03 and 0.00 for the dipole-dipole and the Wenner-Schlumberger arrays, respectively). However, as indicated by negative NSE values, results for the lowest subsolum resistivity contrast (i.e. 1250 ohm.m) are of poor quality (Tables 2 and S2, Figures 6 and S15), especially for the largest depth to bedrock, whose ERT images present strong resistivity artefacts (Figures S11-S12). These poor results can be linked to the reliability of the linear regressions for models with the lowest resistivity contrast. Indeed, as shown in Figures 5 and S14, regression lines cross each other at low apparent resistivity values

and lead to an unsuitable variation of the apparent resistivity. Excluding these models with low resistivity contrasts leads to an increase in the mean NSE values to 0.56 for both arrays, close to those observed for ERT images relying on an ES of 0.5 m (i.e, 0.59 and 0.60 for the dipole-dipole and the Wenner-Schlumberger arrays, respectively, excluding also models with the lowest resistivity contrasts).





Regarding interface delineation, a strong overall improvement is also observed when adding the four surficial interpolated levels to the apparent resistivity datasets using an ES of 2 m (Tables 2 and S2, Figures 6 and S15). The precision and accuracy of the interface depths derived from the second derivative of the resulting ERT images are close to the values obtained from the ERT images based on an ES of 0.5 m. Here again, the improvement of the results is notably smaller in the

case of the lowest resistivity contrast. It is worth noting that the estimates of the largest depth to bedrock are also not satisfactory for subsolum resistivity values of 2500 and 5000 ohm.m, for both arrays in the first case, and the dipole-dipole array solely in the second.

### 3.2 Field case study

The inversion results obtained for the 12 ERT profiles from the Weierbach catchment, with the two standard apparent

resistivity datasets and the upgraded dataset, are presented in Figure 7. Four to twelve iterations were necessary to achieve the inversion process. In each case, an acceptable convergence between the calculated and simulated apparent resistivity data was reached, as indicated by the root mean square misfit error (average: 2.54%, range: 0.94-4.82%) and the $\chi^2$ criteria (average: 1.18, range: 0.39-3.08). For each ERT profile, the median resistivity patterns as a function of depth, as well as the median estimates of solum thickness and depth to hard bedrock derived from the second derivative of ERT images, are

provided in Figure 8.

### 3.2.1 Description of ERT results obtained using an electrode spacing of 0.5 m

As shown in Figure 7-a and Figure 8 (blue thick lines), the variability of resistivity with depth obtained using an ES of 0.5 m correctly reflects the Weierbach catchment subsurface structure. Overall, the observed interpreted resistivity variations are similar for each of the 12 profiles. First, at a depth of less than 0.5 m, the solum has a relatively low resistivity. Then, the

resistivity curves form a sharp peak representing the subsolum, rising on average between 0.5 and 1 m depth and declining between 1 and 1.5 m depth. In the range 1.5-5 m of the fractured bedrock, the interpreted resistivity continues to decline, but the decay is less and less steep as the depth increases. From about 5.0 m depth, resistivity becomes relatively stable.

A clear distinction between the different stages of weathering affecting the regolith is also possible, as revealed by soil pits and drillings. We are able to identify two groups of profiles ( Figures 7-a and 8). Profiles P05, P06, P07, P09, P10 and P12,

located in the north and the west of the catchment, were characterized by overall lower resistivity values for each of the subsurface layers than profiles P01, P02, P03, P04, P08 and P11, which are located on steep slopes and the eastern catchment boundaries. For instance, the resistivity of the solum ranges from about 1500 to 2000 ohm.m for the profiles of the first group, and from around 2000 to 3500 ohm.m for the profiles of the second group. The peak in resistivity characterizing the subsolum reached values between 2000 and 4000 ohm.m for the profiles of the first group. They are much

higher, between 5000 and 11000 ohm.m, for the second group. Finally, for the fresh bedrock pattern, the resistivity is in the order of 100-250 ohm.m and 250-500 ohm for profiles of the first and the second groups, respectively.



Solum thickness and depth to hard bedrock derived from ERT images obtained with an ES of 0.5 m are close to the average estimation values obtained from intrusive investigations (i.e. 0.5 and 1.4 m, respectively; Figure 2) as shown in Figure 8 (blue thin dashed and dot dashed lines) and Table 3, which compiles average values and corresponding standard deviations (averages of all ERT profiles of 0.48 m and 1.78 m, respectively). As we observed in the synthetic modelling exercise in a

similar context (mean depth of 2.01 m for 1 m thick subsolum and using the Wenner-Schlumberger array; Table 1), the depth of the bedrock was nonetheless overestimated. Also note that profiles P01, P02, P03, P04, P08 and P11 exhibit thicker solum overall (average value of 0.57 m), as well as deeper hard bedrock (average value of 2.06 m) than profiles P05, P06, P07, P09, P10 and P12 (average values 0.40 m and 1.49 m, respectively). Again, this observation is in agreement with the divergence observed as a function of the resistivity contrast through the modelling results.

**3.2.2 Comparison of standard and upgraded ERT results obtained using an electrode spacing of 2 m**

For all 12 profiles, the scatter plots in Figure 9 relate the first apparent resistivity acquisition level using an ES of 2 m (horizontal axes) to the first four surficial apparent resistivity levels acquired with an ES of 0.5 m (vertical axes). As with the synthetic modelling results, each of the latter can be derived from the former assuming a linear interpolation. Even if a decreasing accuracy from down to top apparent resistivity levels is noticeable, as indicated by correlation coefficients and

RMSE values, the four linear regressions can be qualified as robust and relevant. From the equations of these linear regressions, the resulting four interpolated levels of surficial apparent resistivity were added to the apparent resistivity datasets using the ES of 2 m to build the upgraded datasets.

NSE values comparing standard ERT images obtained with an ES of 2 m and those using an ES of 0.5 m (Figure 7-ab) clearly suggest an overall decline in geophysical information (mean NSE value of 0.136, range 0.029-0.272), resulting in a

biased picture of the subsurface. Indeed, Figure 7-b and Figure 8 (red thick lines) show that in this case, the vertical resolution is insufficient to assess the solum resistivity pattern correctly. Due to this lack of surficial information, the inversion process converges to a solution where solum and subsolum are almost merged into one single layer of intermediate resistivity. As shown in Figure 10-a, this situation leads to a clear overall overestimation of resistivity values in the solum and a reverse underestimation at subsolum levels. Further deep ERT images appear to still be affected since the comparison

between resistivity in the fractured bedrock also reveals a non-trivial overestimation. Resistivity values in the fresh bedrock are more accurate, as shown by the distribution of resistivity ratios whose centre is very close to 1.

As shown in Figure 7-c and Figure 8 (green thick lines), the enrichment of the apparent resistivity datasets using an ES of 2 m with the four surficial interpolated levels leads to a better solum/subsolum discrimination in the shallow part. This also allowed a more reliable characterization of the subsurface with depth. Indeed, with the exception of the NSE value of profile

P12, which does not vary significantly, all other NSE values (Figure 7-c) indicate that this added surficial constraint is beneficial (i.e. upgraded ERT images obtained with an ES of 2 m better match those using an ES of 0.5 m; mean NSE value 0.353, range 0.2625-0.487). Nonetheless, overall, the inaccuracy remains considerable, as shown by similar dispersion of resistivity ratio distributions, regardless of whether the ERT images were inverted from standard (Figure 10-a) or upgraded




(Figure 10-b) apparent resistivity datasets using an ES of 2 m. We associate this to the small vertical resolution, and to the loss in horizontal resolution. Nonetheless, the overall bias is lower when adding the four interpolated levels of surficial apparent resistivity (i.e. resistivity ratio distribution more centred on the unit value, regardless of the considered regolith horizon considered; Figure 10-b).

The use of standard ERT images obtained with an ES of 2 m to determine solum thickness leads to less accurate and precise values (see average and standard deviation values in Table 3). Most depth estimates tend towards zero because the vertical resolution is inadequate for correctly distinguishing between solum and subsolum layers (Figures 11-a and 12-a). We can nevertheless note that ERT images with higher resistivity contrast lead to an overall better evaluation of the solum thickness (Table 3). This observation is also illustrated in Figure 12-a, where errors have a bimodal distribution with a first peak

centred on -0.5 m and a second peak centred on zero. Furthermore, we obtained less accurate and precise estimates of depth to hard bedrock (Table 3 and the width and skew of the distribution of errors observed in Figure 12-a). As clearly shown in Figure 8, Figure 11-a and Figure 12-a, the depth to bedrock of each profile is strongly overestimated in comparison with depths derived from ERT images using an ES of 0.5 m (mean overestimation of 1.33 m). Overestimation is greater for ERT images with higher resistivity contrasts (Figure 11-a).

As for the accuracy of resistivity distributions, the enrichment of the apparent resistivity datasets using an ES of 2 m with the four surficial interpolated levels is also clearly beneficial for the delineation of interface depths. Indeed, Table 3, Figure 9 and Figure 11-b show that the values obtained for each upgraded ERT profile are closer to those derived from ERT images produced using an ES of 0.5 m for both solum thickness and depth to bedrock. The narrower difference distributions (Figure 12-b), for both solum thickness and depth to bedrock, confirms that results are more precise when adding the four

interpolated levels of surficial apparent resistivity. However, while the distribution is centred on 0 in the case of the soil thickness, it is positively shifted for the depth to bedrock. This overestimation with respect to the depths computed when using an ES of 0.5 m, of mean value of 0.44 m, seems to affect all ERT images, regardless of their resistivity contrast (see Table 3 and Figure 11-b).

## 4 Discussion

**4.1 Inverse solution accuracy issues posed by electrode spacing parameter related choices**

In this study, we investigate a sequence of soil–saprock/saprolite–bedrock. The chosen synthetic three-layered "conductive solum / resistive subsolum / conductive bedrock" structure describes the subsurface of many natural contexts, such as the Weierbach catchment. Through our modelling exercise and the Weierbach catchment case study, we documented the ability and the limitations of ERT to correctly untangle such a typical regolith structure according to the ES parameter. Our results

confirm that the choice of the ES is fundamental for obtaining accurate results, but most importantly it allows us to understand in detail from which ES threshold, why and how the accuracy of the inverted ERT images is affected.





Our results indicate first, for both arrays and whatever the ES retained, that resistivity and thickness contrasts play a key role in the resulting inverted ERT images. In general, for lower resistivity contrasts and shallower structures, the resulting inverted ERT images lead to relatively less well-resolved and fuzzy 3-layer structures. Moreover, mainly for the lowest resistivity contrast, local resistivity artefacts are produced and disturb the accuracy of ERT images. At the opposite end of

the scale, the higher the resistivity contrast and the deeper the structure, the more the ERT images tend towards a sharp, well-defined three-layered structure in our area of interest. However, in this case, for the strongest resistivity contrasts, the interpreted structures shift in depth, resulting in a decrease in ERT image accuracy. These relationships between resistivity contrasts and interpreted resistivity distributions logically affect the interface depths that are extracted from the second derivative of ERT images.

Our study also emphasizes the critical role of the ESs. The impact of these inverse problem effects as a function of the resistivity and thickness contrasts on the accuracy of the geophysical information delivered does indeed largely depend on the ES parameter. While these effects are rather negligible for the smallest ESI, they increasingly deteriorate the accuracy of the ERT images with increasing ES values. More specifically, we observed a threshold effect at an ES value of 0.5 m – as a best compromise to characterize the subsurface. If a larger spacing is retained, the accuracy decreases abruptly in terms of

both resistivity distribution and interface delineation. This finding is valid for both the shallow and deeper horizons of the subsurface. Observations made in the Weierbach catchment fit well this numerical finding. While the use of an ES of 0.5 m gave accurate results, the use of an ES of 2 m produced biased ERT images. In both cases, the ES of 0.5 m corresponds to the thickness of the most surficial layer (i.e. the solum), thus suggesting that the thickness of the solum has to be taken into consideration for the design of ERT surveys.

Indeed, considering the depth-of-investigation of collinear symmetrical four-electrode arrays using the dipole-dipole or the Wenner-Schlumberger arrays (Roy and Apparao, 1971; Barker, 1989), this ES allows a vertical resolution for the shallow parts of the subsurface of about 0.25 m, which corresponds to half the thickness of the uppermost layer. Thus, our results suggest that such a resolution is required. If a larger ES is chosen, the more superficial apparent resistivity measurements are too deep to accurately grasp the surface layer. This oversizing also affects the characterization of deeper layers by causing a

depth-based resistivity bias. This last observation supports previous findings (e.g., Kunetz, 1960; Clément et al., 2009) and allows also a better understanding of some biases observed for deep layers in previous studies in terms of both resistivity distribution and interface depth delineation using derivative methods (Meads et al., 2003; Hirsch et al., 2008; Chambers et al., 2014).

Recently, Chambers et al. (2014) highlighted the very significant challenges in using ERT to detect thin surface layers and

suggested that a reliable resolution of surface layers with a thickness of less than one third of ES should not be expected. This conclusion was based on the interface delineation accuracy, but not on that of the resistivity distribution. Moreover, the use of derivative methods had failed in their case and only isosurface methods gave good results. These methods, which consist of selecting a resistivity threshold value on the basis of intrusive measurements (Chambers et al., 2013; Chambers et al., 2014; Wainwright et al. , 2016), or using statistical analysis of the ERT images (Audebert et al., 2014; Ward et al., 2014),





are indeed less dependent on the sensitivity of ERT images and have shown a greater ability than derivative methods in several cases (Ward et al., 2014; Chambers et al., 2013; Chambers et al., 2014). The success of the application of isosurface methods is however restricted to specific case studies, resulting from the homogeneity of targeted resistivity layers which imply consistent interfaces (Chambers et al., 2013; Chambers et al., 2014; Ward et al., 2014). In other cases, they provide

poor results (Ward et al., 2014; Chambers et al., 2012). Our results are therefore not contradictory with the findings of Chambers et al. (2014).

We ideally recommend using an ES that is close to the thickness of the top subsurface layer in ERT surveys to mirror the architecture and properties of the subsurface correctly. This choice, which is obvious for the characterization of the shallower layer, is also relevant to characterize the subsurface in its entirety – even when solely aiming for the characterization of

deeper layers. However, this recommendation results from one typical subsurface structure and should consequently be transposed to areas of similar characteristics. This means that a generalization of our findings and their interpretation about the inverse solution accuracy problem posed by ES parameter related choices is limited. Nevertheless, the same methodology as followed in this work might be used for other case studies, such as for example the reverse case, i.e. "resistive solum / conductive subsolum / resistive bedrock".

## 4.2 Potential and limitation of the upgrading procedure proposed in this study

The design of an ERT survey consists of a compromise between the need for high resolution for the near surface layer (which would suggest smaller ESs) and the need to cover the area of interest in a reasonable amount of time and to an investigation depth that is deep enough to reconstruct the architecture of the deeper layer (which would give a preference for larger ESs; Chambers et al., 2014).

Eventually, as proposed by Dahlin and Zhou (2004), a quadrupole sequence of apparent resistivity measurements with decreasing vertical resolution and horizontal scanning in depth can reduce operational time without a drastic loss of accuracy. Moreover, in recent years, there has been substantial development of algorithms dedicated to automatically determine non-conventional electrode configurations (Loke et al., 2013). Those algorithms can lead to inverted ERT images whose resolution is superior or equal, respectively with the same or fewer number of measurements, to those using standard

survey designs, as for example Wenner-Schlumberger or dipole-dipole arrays (eg. Stummer et al., 2004; Furman et al, 2004, 2007; Wilkinson et al., 2006, 2012; Loke et al., 2014; Abdullah et al., 2018; Uhleman et al., 2018). In the scope of large-scale ERT surveys, such optimized non-conventional electrode arrays could also help reducing the operational measurement time without reducing the information content. However, setting up the electrodes remains time consuming and the depth of investigation may be insufficient (e.g., ERT device with a limited number of electrodes). If the competing needs to cover the

area of interest are still not reached (i.e. cost and time constraints, adequate depth of investigation), a set-up with larger ESs must be preferred, but the accuracy of the resulting ERT images might be affected by inverse solution reconstruction issues related to the lack of shallow apparent resistivity data as documented within this work. An improvement of these results is nevertheless possible by filling the lack of information in the shallow part of the subsurface. For instance, the deployment of



a fast-moving measurement device (Andrenelli et al., 2013; Guerrero et al., 2016) could be used in parallel to complement the apparent resistivity dataset. Another example, as shown by Clément (2009), is the use of advanced inversion constrained by a priori surficial information to improve the accuracy of ERT images.

Both the synthetic and the Weierbach catchment datasets demonstrated the potential for our novel upgrading procedure to improve the accuracy of large-scale ERT surveys based on large ESs. By adding four surficial apparent resistivity levels to the standard datasets using an ES of 2 m, we improved the vertical resolution solely in the first metre of the subsurface as the depth of investigation curve indicates (Roy and Apparao, 1971; Barker, 1989). However, this focused upgrading led to a better characterization overall in terms of interpreted resistivity distribution and derived interface depths, for both the shallow and deeper horizons of the subsurface. It was this low number of additional data points that improved the solum characterization and its transition with the subsolum, which was missing in the standard apparent resistivity datasets.

The main constraint of the proposed upgrading procedure is that it is only applicable if the shallower layer is relatively homogeneous in terms of resistivity and thickness, which was the case for the synthetic models used. It is indeed this homogeneity that induces the good correlation between the surficial apparent resistivity levels and a deeper level. In the case of the Weierbach catchment, the solum is relatively homogeneous, as indicated by the point-scale investigations available and the 12 plot scale ERT profiles whose locations were distributed between locations with different geomorphological characteristics. It is important to note that local inconsistencies are expected in places where the shallower part of the subsurface will not satisfy the overall solum homogeneity criteria. For instance, in the riparian zone, where solum and subsolum have been eroded, at forest roads, where the soil has been extensively modified (road cut, ballast), or in grasslands surrounding the catchment, where the soil does not have the same characteristics as in the forest zone, the application of the method would most probably lead to erroneous results by inducing false inverted surficial resistivity layers.

A second limitation of the proposed method is pointed out by the synthetic modelling results. The proposed approach fails and even leads to worse results if there is a low resistivity contrast between layers. Indeed, we were able to show that linear regressions leading to the interpolated levels of surficial apparent resistivity crossed each other in this case and led to an unsuitable variation pattern in the apparent resistivity (Figure 6). Ultimately, this causes the formation of false resistivity layers by the inversion process (Figures S11 and S12 in the supplementary material). This problem could be solved to some extent by constraining the linear regressions with respect to each other, so that they do not cross. Other improvements of the method can be anticipated, such as a weighting procedure for the inversion of the interpolated levels of surficial apparent resistivity levels, depending on how well they correlate.

Here again, it is worth recalling that our findings and their interpretation result from one typical subsurface structure. Extra-work is needed to strengthen, and eventually adapt, our upgrading approach to a more general regolith pattern. We especially recommend assessing the proposed methodology for the reverse case "resistive solum / conductive subsolum / resistive bedrock". For instance, a set of three-layered "resistive / conductive / resistive" synthetic models might be explored to confirm/infirm the linear regression logic we highlight in this study.





### 4.3 From the Weierbach catchment perspective and beyond

The ideal design of an ERT survey exploring the architecture and properties of the Weierbach catchment's regolith should rely on an ES of 0.5 m. However in this configuration, a catchment-scale ERT survey appears totally unrealistic, due to obvious inherent time and cost constraints. Despite their narrow depth of investigation, the use of a fast-moving

measurement device might have been a solution to speed up the survey in open landscapes such as grassland or cropland (e.g. Andrenelli et al., 2013; Guerrero et al., 2016), but their deployment in forested areas remains equally cumbersome and time-consuming.

Other geophysical methods exist that might be more efficient than ERT to explore the regolith over large areas (Binley et al., 2015; Parsekian et al., 2015; Fan et al., 2020). For instance, ground penetrating radar (GPR) allows usually a higher spatial

resolution and data collection rate. In several studies, GPR has allowed to accurately delineate interfaces of several relevant structures in the critical zone (e.g. Carrière et al., 2013; Hare et al., 2017; Guo et al., 2020; Šamonil et al., 2020).  However, as GPR requires sliding the instrument on the ground, its use is much more ticklish and time-consuming over long distances in forests than in open areas, such as grassland or cropland. Furthermore, GPR surveys were done in the close vicinity of the Weierbach catchment. The structural analysis only revealed soil layering, but it did not show the depth to bedrock due to

chaotic reflection patterns, which are common for this type of geologic setting (Jackisch et al., 2017; Allroggen et al., 2020). The electromagnetic induction (EMI) method could also have been another way to quickly characterize the shallow subsurface at large scale. Indeed, unlike GPR or ERT, EMI systems do not require a direct coupling with the ground, which allows much faster acquisitions, even in forested areas. Despite their limited spatial resolution and depth of investigation, multi-depth EMI devices have been used in several studies for the characterization of subsurface structures and properties

(e.g. Brosten et al., 2011; Saey et al., 2012; Rejiba et al., 2018; Simon et al., 2020). A field test using a multi-frequency domain EMI device (Profiler EMP-400, GSSI) was done in the Weierbach catchment. Unfortunately, the results were inconclusive, as contrasting shallow patterns observed with ERT were in fact not distinguishable due to the overall too electrically resistive nature of the subsurface.

Hence, the upgrading procedure proposed in this study is particularly interesting in the context of the Weierbach catchment.

Through our study, we demonstrated that applying this new approach to the existing catchment-wide ERT dataset measured with an ES of 2 m contributes to an improved characterization of the regolith. From a hydrological perspective, the deployment of the upgrading procedure at catchment scale is promising as it could bring new insights in terms of hydrological process understanding and modelling. Indeed, in the past years several investigations have pointed out the critical role of the Weierbach subsurface in its hydrological functioning (Pfister et al., 2010; Fenicia et al., 2014; Wrede et

al., 2015; Martínez-Carreras et al., 2015; Martínez-Carreras et al., 2016; Pfister et al., 2017; Scaini et al., 2017, 2018). Recently, using a 3D integrated hydrological surface-subsurface modelling approach, Glaser et al. (2019) were able to bring further evidences that the multi-layered nature - with contrasting hydraulic properties and effective conductivities - of the Weierbach regolith is responsible for the main processes controlling the hydrometric response in the catchment, i.e. fast





vertical flow in the unsaturated zone combined with connected fast lateral subsurface flow. However, although the subsurface plays a key role in the hydrological functioning of the Weierbach catchment, its spatial variability has been taken into account only minimally for the moment. In the most recent hydrological model of the catchment for example, the spatial variability of the subsurface is only considered in the stream valleys, where solum and subsolum were eroded and the

outcropping fractured bedrock is overlain with organic material (elsewhere in the catchment, the subsurface structure and properties was parameterized homogenously; Glaser et al., 2020).

Yet, the intrusive point-scale investigation suggests a potential significant spatial variability of the subsurface hydraulic properties (in close relation with the observed soil clay content, subsolum slate fragment size and bedrock weathering heterogeneity; see 2.1.1) that which might be derived from the catchment-wide ERT survey as suggested in this study (see

3.2.1). Moreover, as highlighted by Loritz et al. (2017) in a nearby catchment with the same regolith structure, the bedrock topography plays a significant role in the interplay of water flow and storage in our study area. Using a physically-based hillslope modelling approach, they showed that a model with surface-parallel bedrock topographies performed considerably worse in matching streamflow than a model including a bedrock topography. Furthermore, in their model, the topography of the bedrock was successfully constrained with an ERT survey using an ES of 0.5 m (Loritz et al., 2017), thus also

underlining the added-value that can be expected from the upgrading approach proposed in this study.

The presented elements suggest that the application of the upgrading procedure to the catchment-wide ERT survey dataset relying on an ES of 2 m constitutes a promising added-value that might improve model realism of the Weierbach catchment (Clark et al., 2017). We further expect that our novel approach may also be transferable to catchments with similar characteristics, like forested catchments with similar bedrock geology (e.g. Bellot and Ortiz de Urbina, 2008; Hübner et al.,

2015). Specifically, the regolith of the Weierbach catchment is representative of the slate regolith which covers a large part of the Rhenish Massif (Moragues-Quiroga et al., 2017). Hence, we anticipate that the proposed protocol could be used at several regions of this large central European geological area that extends from Luxembourg, through Belgium, France and Germany (Sauer and Felix-Henningsen, 2006).

## 5 Summary and conclusions

An accurate knowledge of regolith is needed in catchment studies to better understand and predict subsurface water flow paths, transit times and storage volumes. However, the characterization of the subsurface is stymied by the invasive and "point-scale" characters of traditional investigation techniques, essentially because of time and cost constraints. ERT is one of the geophysical tools at hand to overcome this limitation. This technique is now commonly used in the critical zone to disentangle regolith properties and architecture, but its use should go hand in hand with a suitable resolution of ERT images.

In this paper, we discuss the importance of ESs on the quality of ERT images to adequately mirror subsurface resistivity distributions and accurately delineate interfaces. To this end, we investigated a synthetic "conductive / resistive / conductive" three-layered sequence of soil–saprock/saprolite–bedrock, which mirrors the subsurface of many natural contexts, in



combination with the Weierbach catchment field dataset, as a reference case study. Inversion results obtained for different ESs were compared in terms of resistivity distribution accuracy. We also inferred interface depths from each ERT image using a derivative method and evaluated their accuracy.

Our results highlight the need to use an adapted vertical resolution to best mirror the structure of the subsurface. More
specifically, we document the inverse solution reconstruction issues related to the lack of shallow apparent resistivity data induced by the use of an oversized ES. We found out that the thickness of the most superficial layer must be taken into consideration when choosing the ES. Specifically, we demonstrated that the best compromise consists of using an ES close to the thickness of the subsurface top layer. If a larger ES is retained, the accuracy of the results decreases rapidly in terms of both resistivity distribution and interface delineation. This choice, which is obvious for the characterization of the shallower
layer, is also relevant to characterize the subsurface in its entirety – even when solely aiming for the characterization of deeper layers. For instance, our observations obviously support previous findings and confirm that oversizing the ESs not only leads to an inappropriate vertical resolution for the delineation of thin surface layers, but that it also affects the outlining of deeper layers. In particular, we demonstrated that an oversized ES leads to overestimations of the depth to bedrock and that this overestimation is even more important for subsurface structures with high resistivity contrast.

To overcome this limitation, we propose adding interpolated levels of surficial apparent resistivity based on a limited number of ERT profiles with a small ES that satisfies the thickness of the top subsurface layer. We show that our protocol significantly improves the accuracy of ERT profiles based on large ESs, provided that the top layer has a rather constant thickness and resistivity, such as the solum in the Weierbach catchment. Our results demonstrated that this upgrading procedure is promising for carrying out large-scale surveys in a cost-effective and more robust way, for instance to feed
hydrological models with subsurface structure depths and properties at catchment scale. However, our findings and their interpretation result from one typical regolith logic and extra-work is needed to strengthen, and eventually adapt, our upgrading approach to a more general regolith pattern. We especially recommend assessing the proposed methodology for the reverse case "resistive solum / conductive subsolum / resistive bedrock".

**Acknowledgments**

This work was funded by the Luxembourg National Research Fund (FNR) as part of the SOWAT (FNR/CORE/C10/SR/799842/SOWAT) and CAOS2 (FNR/INTER/DFG/14/02/CAOS2) projects. We would like to thank Cyrille Tailliez and Jean-Francois Iffly for their help during the field campaigns. We also express gratitude to T. Günther for providing BERT2 software and supportive guidance. Data used to generate the resistivity models are archived by the Luxembourg Institute of Science and Technology, and are available by contacting the corresponding author.



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



# Figures

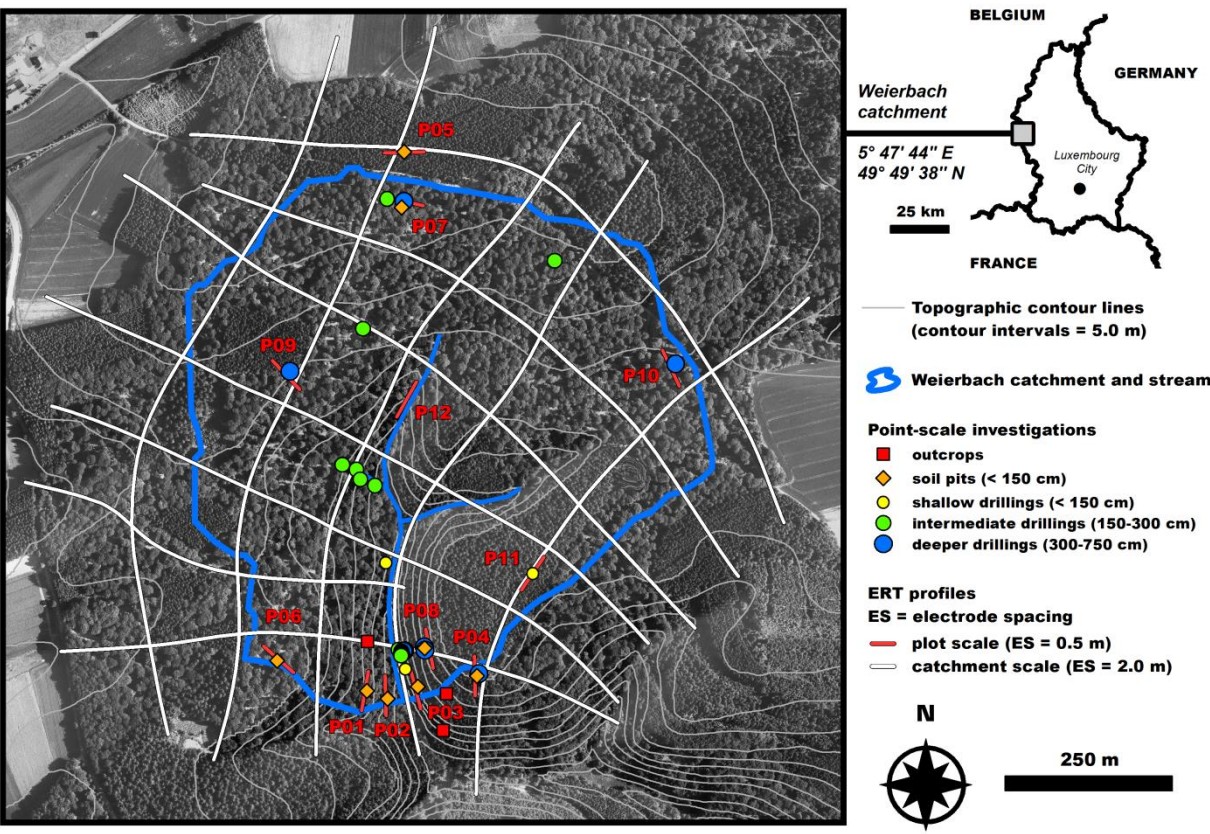

**Figure 1:** Detailed map of topography and investigations made in the Weierbach catchment (background aerial photography from
*Administration du Cadastre et de la Topographie, Luxembourg*).



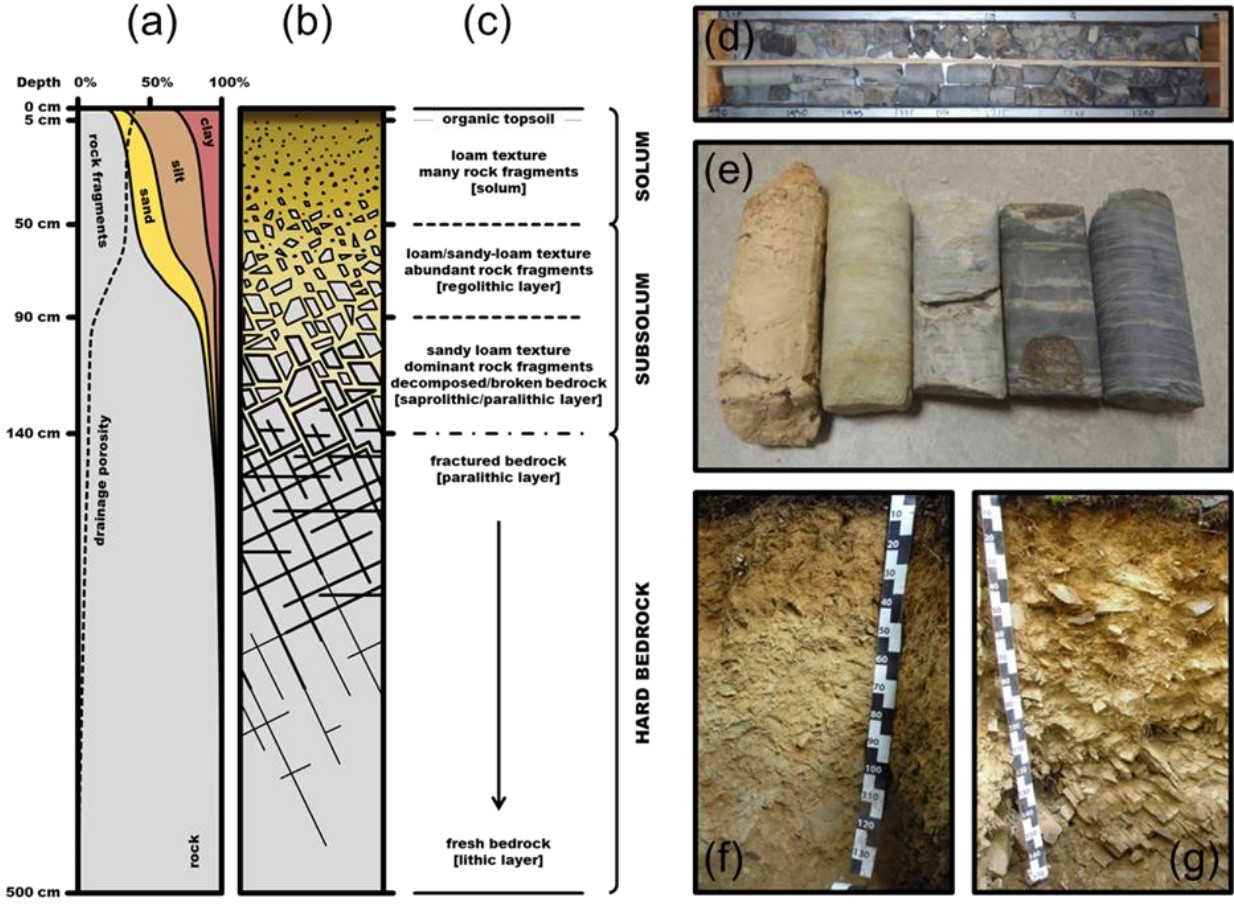

**Figure 2:** Synthesis scheme of the regolith in the Weierbach catchment (with (a) variations with depth of average drainage porosity and rock, sand, silt and clay contents, (b) sketch of the regolith and (c) description of regolith layers) and pictures of some "point-scale" investigation spots (with (d) cores from drilling carried out next to plot scale ERT profile P04, (e) various aspects of the top of the substratum as revealed in deeper drillings, (f) soil pit dug next to plot scale ERT profile P08 and (d) soil pit dug next to plot scale ERT profile P07).

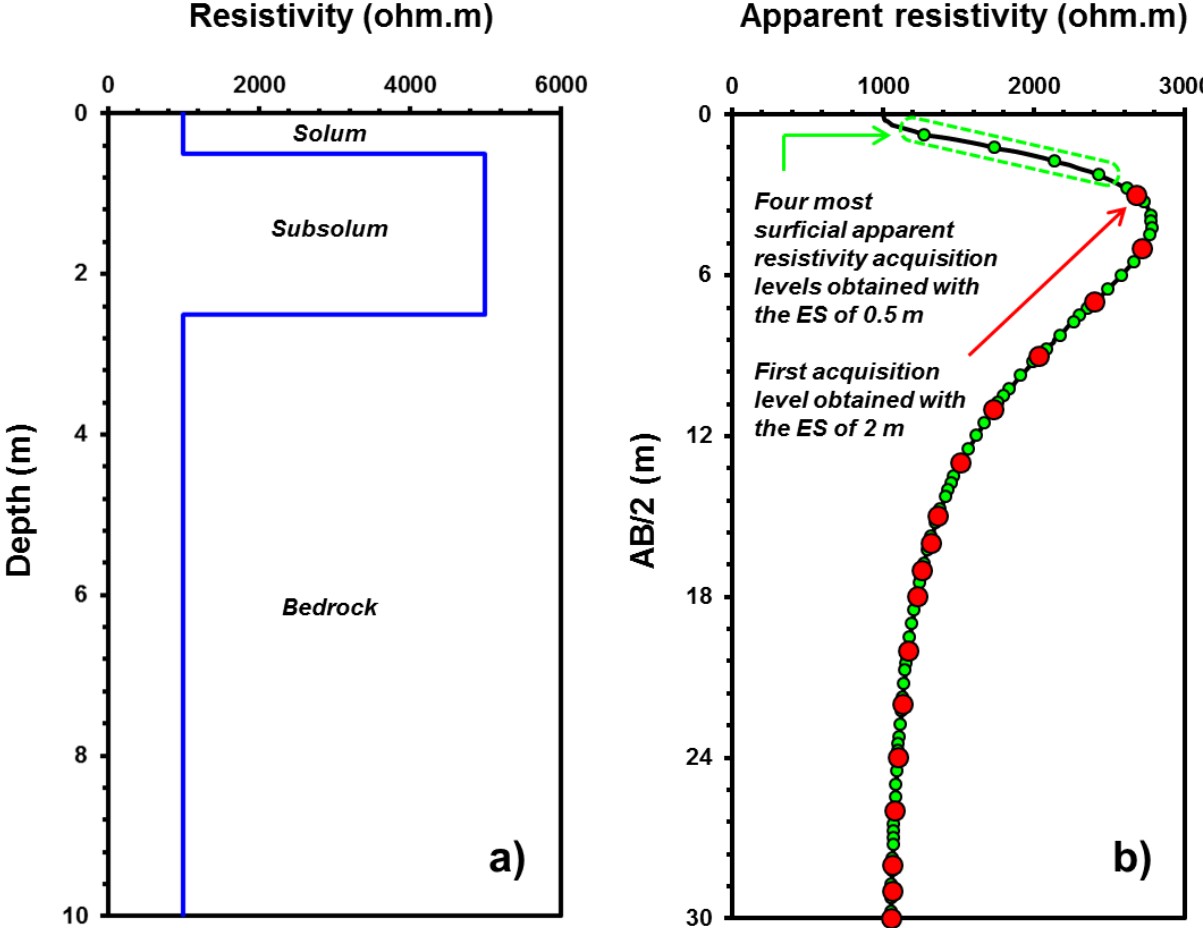

**Figure 3:** Illustration of the lack of shallow apparent resistivity data posed by the ES parameter related choice exemplified with a) a 1D geoelectric model (blue line) and b) corresponding Wenner-Schlumberger array apparent resistivity curve (black line) and measurements (green and red dots using respectively an ES of 0.5 m and 2 m). AB/2 stands for half of the external electrode separations.





**Figure 4:** Nash–Sutcliffe model efficiency coefficient and mean interface depths resulting from the inversion of the 25 synthetic apparent resistivity models using the Wenner-Schlumberger array with the five different ESs. In plots showing the estimated interface depths, thick black lines indicate the expected values.


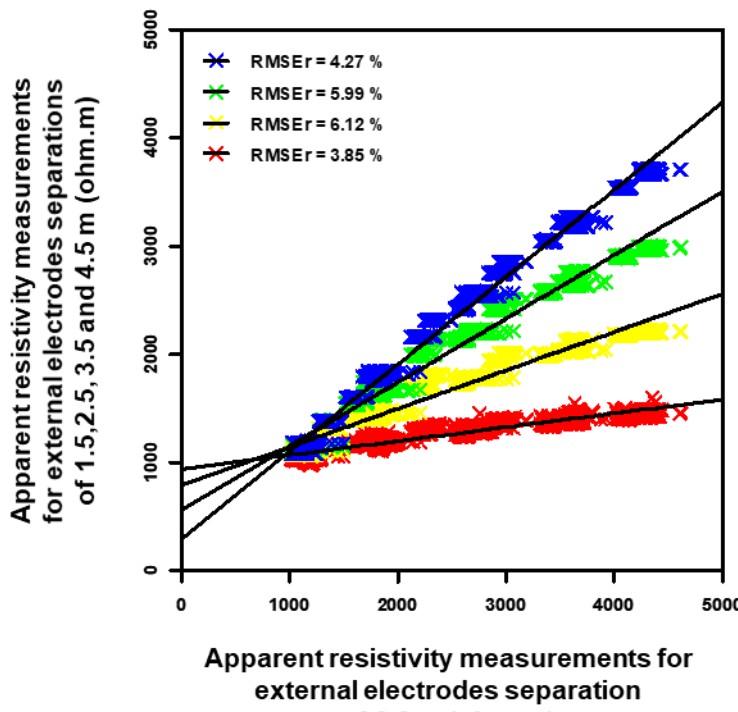

**Figure 5:** Scatter plots showing the first apparent resistivity level for an ES of 2 m (external electrodes spacing of 6 m) versus the four surficial apparent resistivity levels for an ES of 0.5 m with external electrodes separations of 1.5 (red crosses), 2.5 (yellow crosses), 3.5 (green crosses) and 4.5 m (blue crosses) using the dipole-dipole (DD) and the Wenner-Schlumberger (WS) arrays for the 25 synthetic resistivity models. The linear regressions correspond to the thick black lines and their accuracy is indicated by the root mean square relative error (RMSEr).

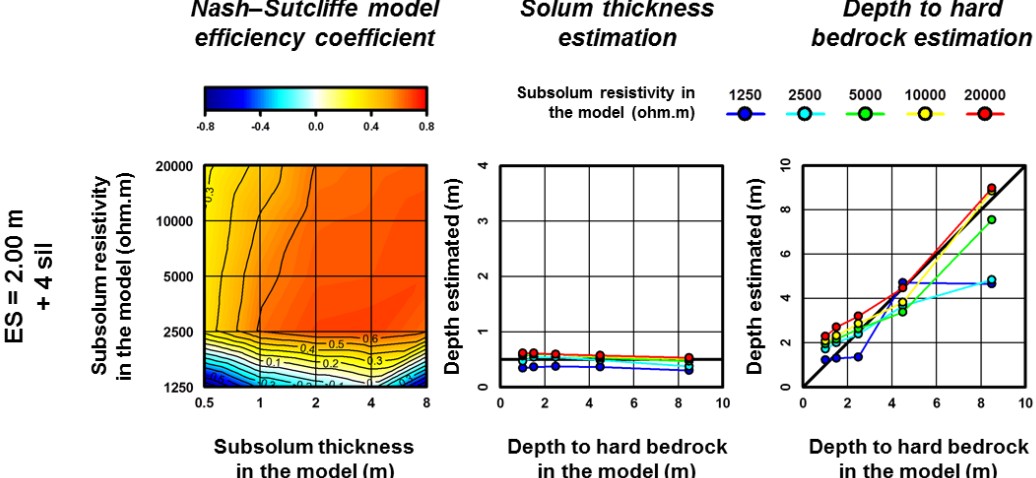

**Figure 6:** Nash–Sutcliffe model efficiency coefficient and mean interface depths resulting from the inversion of the 25 synthetic apparent resistivity models using the dipole-dipole (DD) and the Wenner-Schlumberger (WS) arrays with an ES of 2 m and upgraded with the four interpolated levels of surficial apparent resistivity (sil stands for surficial interpolated levels). In plots showing the estimated interface depths, thick black lines indicate the expected values.



**Figure 7:** Inversion results obtained for the 12 plot scale ERT profiles measured in the Weierbach catchment using an ES of 0.5 m (a) or 2 m without (b) or with the four interpolated levels of surficial apparent resistivity (c). Nash–Sutcliffe model efficiency (NSE) values were added to each ERT image relying on an ES of 2 m using ERT images obtained with an ES of 0.5 m as references.







**Figure 8:** Median resistivity as a function of depth for the 12 plot scale ERT profiles measured in the Weierbach catchment using an ES of 0.5 m (blue thick curves) or 2 m without (red thick curves) or with the four interpolated levels of surficial apparent resistivity (green thick curves). Median interface depths derived from the second derivative of ERT images are indicated by thin dashed lines for solum thickness and thin dot dashed lines for depth to hard bedrock (coloured in the same way as the median resistivity curves).



**Figure 9:** Scatter plots showing the first apparent resistivity level for an ES of 2 m (external electrodes spacing of 6 m) versus the first four surficial apparent resistivity levels for an ES of 0.5 m with external electrodes separations of 1.5 (red crosses), 2.5 (yellow crosses), 3.5 (green crosses) and 4.5 m (blue crosses) for the 12 plot scale ERT profiles measured in the Weierbach catchment. The linear regressions correspond to the thick black lines and their accuracy is indicated by the coefficient of determination ($R^2$), the root mean square error (RMSE) and the root mean square relative error (RMSEr).





**Figure 10:** Distribution of the ratios calculated between the inverted resistivities obtained using an ES of 2 m without (a) or with the four interpolated levels of surficial apparent resistivity (b) and those obtained using an ES of 0.5 m considering the overall 12 plot scale ERT profiles measured in the Weierbach catchment and discretized by relevant depth horizons. A lognormal distribution, whose centre is indicated by a vertical line, has been fitted for each histogram; the more the distribution is centred and narrowed on the unit ratio (vertical blue lines), the better the adequacy with ERT images using an ES of 0.5 m.



**Figure 11:** Coloured density scatter plots (red – high density to blue – low density) showing solum thickness and depth to hard bedrock derived from the 12 plot scale ERT profiles measured in the Weierbach catchment using an ES of 0.5 m versus those using an ES of 2 m without (a) or with (b) the four interpolated levels of surficial apparent resistivity (b). Median values and interpercentile ranges of 10-90% of the interface depth of each ERT profile are shown by black dots and thin vertical and horizontal bars. Envelopes defined by thin dashed black contours encompass 80% of individual pairs of values.





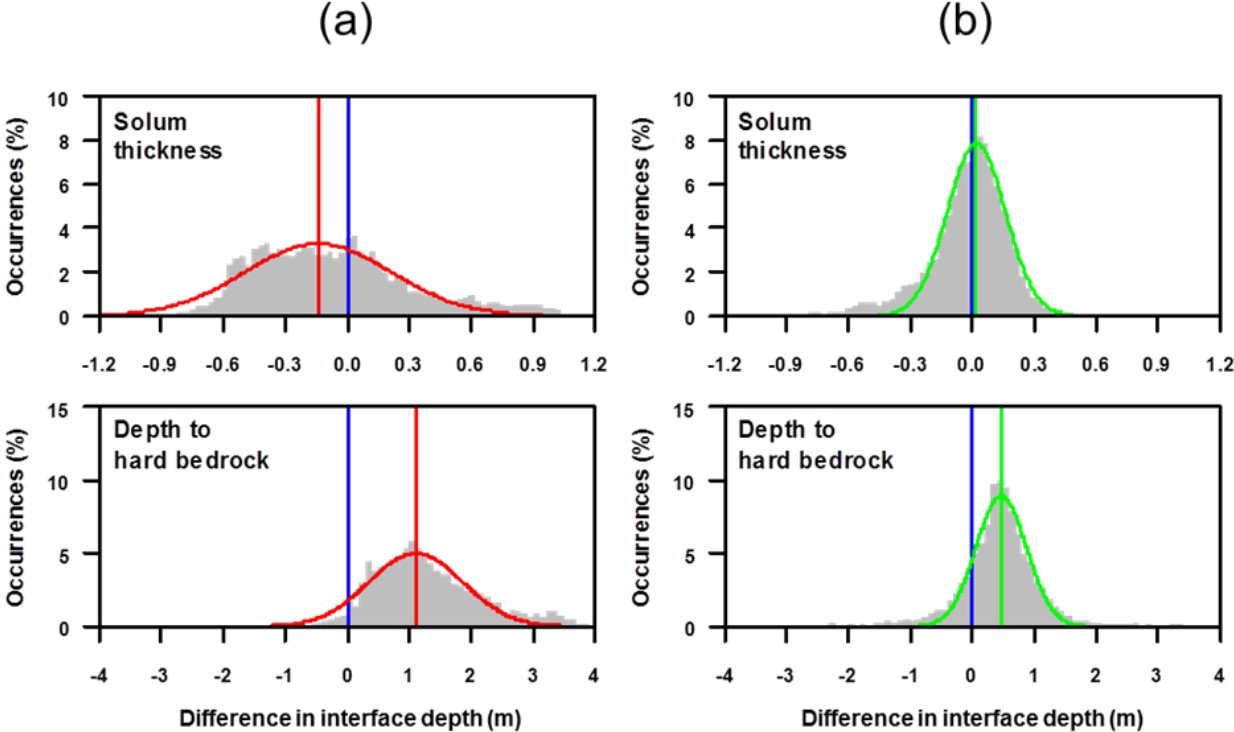

**Figure 12:** Distribution of differences between interface depths obtained using an ES of 2 m without (a) or with the four interpolated levels of surficial apparent resistivity (b) and those obtained using an ES of 0.5 m considering the overall 12 plot scale ERT profiles measured in the Weierbach catchment. A normal distribution, whose centre is indicated by a vertical line, has been fitted for each histogram; the more the distribution is centred and narrowed on the zero value (vertical blue lines), the better the adequacy with interfaces derived from ERT images using an ES of 0.5 m.




**Tables**

**Table 1:** Nash–Sutcliffe model efficiency coefficient (NSE) and interface depths (avg ± sd, average ± standard deviation in m; an italic value specifies that the interface delineation results from the merging of several second derivative zero contours) resulting from the inversion of the 25 synthetic apparent resistivity models (Tss, subsolum thickness in m; Rss, subsolum resistivity in ohm.m) using the Wenner-Schlumberger array with the 5 different ESs.

| | | ES = 0.25 m | | | ES = 0.50 m | | | ES = 1.00 m | | | ES = 2.00 m | | | ES = 4.00 m | | |
|---|---|---|---|---|---|---|---|---|---|---|---|---|---|---|---|---|---|
| | | | Solum depth | Depth to bedrock | | Solum depth | Depth to bedrock | | Solum depth | Depth to bedrock | | Solum depth | Depth to bedrock | | Solum depth | Depth to bedrock |
| Tss | Rss | NSE | avg ± sd | avg ± sd | NSE | avg ± sd | avg ± sd | NSE | avg ± sd | avg ± sd | NSE | avg ± sd | avg ± sd | NSE | avg ± sd | avg ± sd |
| 0.5 | 1250 | 0.16 | 0.47 ± 0.06 | 1.30 ± 0.12 | 0.21 | 0.37 ± 0.05 | 1.40 ± 0.16 | 0.23 | 0.44 ± 0.09 | 1.69 ± 0.21 | -0.24 | *0.28 ± 0.20* | 1.49 ± 0.66 | 0.06 | *0.28 ± 0.39* | *2.01 ± 1.33* |
| 0.5 | 2500 | 0.48 | 0.47 ± 0.04 | 1.37 ± 0.14 | 0.43 | 0.39 ± 0.04 | 1.45 ± 0.14 | 0.17 | 0.51 ± 0.04 | 1.75 ± 0.17 | -0.29 | *0.21 ± 0.13* | 1.60 ± 0.29 | -0.05 | *0.24 ± 0.32* | *2.43 ± 0.49* |
| 0.5 | 5000 | 0.49 | 0.48 ± 0.04 | 1.50 ± 0.15 | 0.45 | 0.44 ± 0.04 | 1.59 ± 0.16 | 0.21 | 0.59 ± 0.04 | 1.93 ± 0.19 | -0.15 | *0.28 ± 0.20* | 2.28 ± 0.20 | -0.09 | *0.18 ± 0.21* | 2.43 ± 0.32 |
| 0.5 | 10000 | 0.45 | 0.50 ± 0.04 | 1.67 ± 0.19 | 0.40 | 0.51 ± 0.04 | 1.80 ± 0.20 | 0.20 | 0.70 ± 0.05 | 2.22 ± 0.21 | -0.01 | 0.98 ± 0.06 | 2.93 ± 0.17 | -0.07 | *0.25 ± 0.17* | 2.69 ± 0.19 |
| 0.5 | 20000 | 0.37 | 0.52 ± 0.04 | 1.91 ± 0.23 | 0.31 | 0.58 ± 0.04 | 2.10 ± 0.24 | 0.18 | 0.83 ± 0.06 | 2.62 ± 0.23 | 0.12 | 1.16 ± 0.08 | 3.40 ± 0.20 | 0.07 | *0.52 ± 0.23* | 3.70 ± 0.30 |
| 1 | 1250 | 0.51 | 0.53 ± 0.05 | 1.59 ± 0.22 | 0.40 | 0.55 ± 0.08 | 1.68 ± 0.19 | 0.48 | 0.53 ± 0.09 | 1.87 ± 0.20 | -0.47 | *0.29 ± 0.29* | 1.78 ± 0.64 | -0.32 | *0.33 ± 0.31* | 2.15 ± 0.89 |
| 1 | 2500 | 0.69 | 0.53 ± 0.03 | 1.69 ± 0.19 | 0.67 | 0.51 ± 0.04 | 1.79 ± 0.19 | 0.47 | 0.60 ± 0.04 | 2.04 ± 0.19 | -0.27 | 0.68 ± 0.14 | 2.65 ± 0.11 | -0.29 | *0.26 ± 0.21* | 2.42 ± 0.32 |
| 1 | 5000 | 0.69 | 0.53 ± 0.03 | 1.79 ± 0.20 | 0.65 | 0.54 ± 0.04 | 1.92 ± 0.22 | 0.38 | 0.70 ± 0.05 | 2.27 ± 0.22 | -0.08 | 1.01 ± 0.07 | 2.99 ± 0.17 | -0.21 | *0.26 ± 0.18* | 2.73 ± 0.17 |
| 1 | 10000 | 0.65 | 0.54 ± 0.04 | 1.98 ± 0.23 | 0.55 | 0.59 ± 0.04 | 2.16 ± 0.25 | 0.26 | 0.84 ± 0.06 | 2.65 ± 0.22 | 0.05 | 1.17 ± 0.08 | 3.44 ± 0.20 | -0.02 | *0.55 ± 0.23* | 3.77 ± 0.31 |
| 1 | 20000 | 0.56 | 0.55 ± 0.05 | 2.25 ± 0.26 | 0.43 | 0.65 ± 0.05 | 2.50 ± 0.30 | 0.17 | 0.98 ± 0.07 | 3.00 ± 0.24 | 0.15 | 1.31 ± 0.11 | 4.02 ± 0.16 | 0.10 | 1.48 ± 0.19 | 5.25 ± 0.40 |
| 2 | 1250 | 0.50 | 0.58 ± 0.07 | 1.98 ± 0.38 | 0.31 | *0.64 ± 0.07* | 1.90 ± 0.27 | 0.53 | 0.56 ± 0.10 | 2.08 ± 0.27 | -0.38 | *0.69 ± 0.44* | 2.40 ± 0.88 | -0.47 | *0.26 ± 0.30* | 2.39 ± 0.60 |
| 2 | 2500 | 0.69 | 0.54 ± 0.04 | 2.20 ± 0.29 | 0.67 | 0.57 ± 0.05 | 2.19 ± 0.26 | 0.53 | 0.71 ± 0.06 | 2.49 ± 0.25 | -0.02 | 1.08 ± 0.09 | 3.23 ± 0.19 | -0.29 | *0.35 ± 0.20* | 3.08 ± 0.21 |
| 2 | 5000 | 0.68 | 0.55 ± 0.04 | 2.32 ± 0.28 | 0.65 | 0.59 ± 0.05 | 2.40 ± 0.28 | 0.47 | 0.82 ± 0.07 | 2.80 ± 0.22 | 0.13 | 1.20 ± 0.08 | 3.60 ± 0.21 | -0.10 | *0.64 ± 0.27* | 4.21 ± 0.34 |
| 2 | 10000 | 0.65 | 0.55 ± 0.05 | 2.50 ± 0.28 | 0.61 | 0.63 ± 0.05 | 2.68 ± 0.26 | 0.39 | 0.96 ± 0.06 | 3.09 ± 0.24 | 0.19 | 1.34 ± 0.11 | 4.12 ± 0.17 | 0.01 | 1.61 ± 0.19 | 5.46 ± 0.42 |
| 2 | 20000 | 0.62 | 0.57 ± 0.06 | 2.79 ± 0.23 | 0.57 | 0.68 ± 0.06 | 2.98 ± 0.24 | 0.31 | 1.09 ± 0.08 | 3.58 ± 0.27 | 0.10 | 1.64 ± 0.15 | 4.76 ± 0.30 | 0.02 | 2.41 ± 0.13 | 6.93 ± 0.44 |
| 4 | 1250 | 0.63 | 0.48 ± 0.06 | 4.34 ± 0.18 | 0.51 | *0.43 ± 0.15* | 4.35 ± 0.24 | 0.17 | *0.91 ± 0.37* | 3.50 ± 0.24 | -0.07 | 1.06 ± 0.12 | 3.20 ± 0.20 | -0.59 | *0.35 ± 0.33* | *2.71 ± 0.97* |
| 4 | 2500 | 0.78 | 0.51 ± 0.04 | 4.68 ± 0.20 | 0.73 | 0.50 ± 0.05 | 4.70 ± 0.28 | 0.53 | 0.75 ± 0.08 | 3.56 ± 0.32 | 0.19 | 1.21 ± 0.10 | 3.99 ± 0.18 | -0.10 | 1.36 ± 0.33 | 5.38 ± 0.55 |
| 4 | 5000 | 0.76 | 0.52 ± 0.04 | 4.91 ± 0.23 | 0.70 | 0.53 ± 0.06 | 4.58 ± 0.41 | 0.51 | 0.85 ± 0.08 | 3.65 ± 0.29 | 0.27 | 1.36 ± 0.12 | 4.45 ± 0.25 | -0.02 | 1.96 ± 0.21 | 6.31 ± 0.57 |
| 4 | 10000 | 0.70 | 0.53 ± 0.05 | 4.67 ± 0.47 | 0.64 | 0.58 ± 0.06 | 4.15 ± 0.55 | 0.41 | 1.01 ± 0.07 | 3.87 ± 0.24 | 0.19 | 1.66 ± 0.16 | 5.07 ± 0.32 | -0.03 | 2.53 ± 0.14 | 7.29 ± 0.39 |
| 4 | 20000 | 0.61 | 0.54 ± 0.06 | *3.89 ± 0.49* | 0.57 | 0.65 ± 0.06 | 3.99 ± 0.42 | 0.31 | 1.16 ± 0.10 | 4.33 ± 0.29 | 0.04 | 1.99 ± 0.18 | 6.14 ± 0.49 | -0.09 | 2.96 ± 0.17 | 8.38 ± 0.31 |
| 8 | 1250 | 0.65 | 0.46 ± 0.07 | 9.19 ± 0.33 | 0.53 | *0.34 ± 0.11* | 9.52 ± 0.24 | -0.19 | *1.36 ± 0.25* | 10.8 ± 0.30 | -0.24 | 1.15 ± 0.23 | 3.68 ± 0.33 | -0.34 | 1.55 ± 0.59 | 5.57 ± 0.77 |
| 8 | 2500 | 0.78 | 0.51 ± 0.05 | 8.33 ± 0.40 | 0.77 | 0.48 ± 0.05 | 8.63 ± 0.25 | 0.48 | 0.86 ± 0.10 | 9.35 ± 0.27 | 0.23 | 1.30 ± 0.17 | 4.26 ± 0.28 | -0.07 | 2.11 ± 0.39 | 7.33 ± 0.47 |
| 8 | 5000 | 0.75 | 0.52 ± 0.05 | *9.21 ± 0.93* | 0.74 | 0.53 ± 0.05 | 8.67 ± 0.26 | 0.50 | 0.90 ± 0.07 | 9.47 ± 0.31 | 0.22 | 1.50 ± 0.15 | 5.04 ± 0.31 | -0.02 | 2.63 ± 0.17 | 8.41 ± 0.35 |
| 8 | 10000 | 0.73 | 0.54 ± 0.05 | 9.08 ± 0.51 | 0.71 | 0.58 ± 0.06 | 9.20 ± 0.37 | 0.46 | 0.98 ± 0.07 | 9.83 ± 0.53 | 0.19 | 1.77 ± 0.15 | 6.30 ± 0.59 | -0.02 | 2.96 ± 0.16 | 9.08 ± 0.38 |
| 8 | 20000 | 0.69 | 0.54 ± 0.06 | 9.66 ± 0.36 | 0.64 | 0.62 ± 0.06 | 10.1 ± 0.50 | 0.40 | 1.07 ± 0.09 | 8.92 ± 1.08 | 0.14 | 2.04 ± 0.21 | 7.20 ± 0.48 | -0.02 | 3.32 ± 0.18 | 9.93 ± 0.50 |





**Table 2:** Nash–Sutcliffe model efficiency coefficient (NSE) and interface depths (avg ± sd, average ± standard deviation in m; an italic value specifies that the interface delineation results from the merging of several second derivative zero contours) resulting from the inversion of the 25 synthetic apparent resistivity models (Tss, subsolum thickness in m; Rss, subsolum resistivity in ohm.m) using the Wenner-Schlumberger (WS) array with the ES of 2 m upgraded with the four interpolated levels of surficial apparent resistivity (sil stands for surficial interpolated levels).

ES = 2.00 m + 4sil

| Tss | Rss | NSE | Solum depth | | | Depth to bedrock | | |
|-----|------|-------|------|---|------|------|---|------|
| | | | avg | ± | sd | avg | ± | sd |
| 0.5 | 1250 | -1.16 | 0.35 | ± | 0.05 | 1.23 | ± | 0.11 |
| 0.5 | 2500 | 0.34 | 0.47 | ± | 0.05 | 1.73 | ± | 0.14 |
| 0.5 | 5000 | 0.33 | 0.57 | ± | 0.04 | 1.95 | ± | 0.20 |
| 0.5 | 10000 | 0.30 | 0.59 | ± | 0.04 | 2.10 | ± | 0.23 |
| 0.5 | 20000 | 0.27 | 0.62 | ± | 0.05 | 2.29 | ± | 0.27 |
| 1 | 1250 | -0.38 | 0.37 | ± | 0.04 | 1.31 | ± | 0.15 |
| 1 | 2500 | 0.62 | 0.54 | ± | 0.05 | 2.02 | ± | 0.19 |
| 1 | 5000 | 0.57 | 0.60 | ± | 0.04 | 2.17 | ± | 0.23 |
| 1 | 10000 | 0.51 | 0.61 | ± | 0.05 | 2.34 | ± | 0.27 |
| 1 | 20000 | 0.43 | 0.62 | ± | 0.05 | 2.71 | ± | 0.26 |
| 2 | 1250 | -0.20 | 0.38 | ± | 0.04 | 1.37 | ± | 0.18 |
| 2 | 2500 | 0.69 | 0.54 | ± | 0.06 | 2.40 | ± | 0.27 |
| 2 | 5000 | 0.65 | 0.60 | ± | 0.05 | 2.63 | ± | 0.25 |
| 2 | 10000 | 0.64 | 0.60 | ± | 0.05 | 2.88 | ± | 0.22 |
| 2 | 20000 | 0.61 | 0.59 | ± | 0.06 | *3.21* | ± | *0.23* |
| 4 | 1250 | -0.04 | 0.36 | ± | 0.04 | 4.72 | ± | 0.32 |
| 4 | 2500 | 0.70 | 0.49 | ± | 0.06 | *3.68* | ± | *0.44* |
| 4 | 5000 | 0.66 | 0.54 | ± | 0.06 | *3.38* | ± | *0.28* |
| 4 | 10000 | 0.65 | 0.55 | ± | 0.06 | *3.84* | ± | *0.47* |
| 4 | 20000 | 0.63 | 0.57 | ± | 0.06 | 4.47 | ± | 0.49 |
| 8 | 1250 | -0.76 | 0.30 | ± | 0.06 | 4.65 | ± | 0.19 |
| 8 | 2500 | 0.62 | 0.38 | ± | 0.08 | 4.86 | ± | 0.46 |
| 8 | 5000 | 0.68 | 0.48 | ± | 0.07 | *7.53* | ± | *1.44* |
| 8 | 10000 | 0.68 | 0.51 | ± | 0.07 | 8.85 | ± | 0.58 |
| 8 | 20000 | 0.64 | 0.53 | ± | 0.06 | 8.97 | ± | 0.58 |





**Table 3:** Interface depths (avg ± sd, average ± standard deviation in m) derived from the inversion results obtained for the 12 plot scale ERT profiles measured in the Weierbach catchment using an ES of 0.5 m or 2 m (upgraded or not with the four interpolated levels of surficial apparent resistivity; sil stands for surficial interpolated levels). Mean differences (md) in interface depths with the reference ERT images relying on an ES of 0.5 m were added to the standard and upgraded results obtained with an ES of 2 m.

| | ES = 0.50 m | | | | ES = 2.00 m | | | | | | ES = 2.00 m + 4sil | | | | | |
|---|---|---|---|---|---|---|---|---|---|---|---|---|---|---|---|---|
| | Solum depth | | Depth to bedrock | | Solum depth | | | Depth to bedrock | | | Solum depth | | | Depth to bedrock | | |
| Profile | avg ± sd | | avg ± sd | | avg ± sd | | [ md ] | avg ± sd | | [ md ] | avg ± sd | | [ md ] | avg ± sd | | [ md ] |
| P01 | 0.59 ± 0.19 | | 2.21 ± 0.40 | | 0.70 ± 0.44 | | [0.11] | 3.63 ± 0.59 | | [1.42] | 0.52 ± 0.10 | | [-0.07] | 2.55 ± 0.39 | | [0.34] |
| P02 | 0.54 ± 0.11 | | 2.01 ± 0.46 | | 0.59 ± 0.40 | | [0.05] | 3.44 ± 0.55 | | [1.43] | 0.50 ± 0.08 | | [-0.04] | 2.39 ± 0.32 | | [0.38] |
| P03 | 0.69 ± 0.31 | | 2.17 ± 0.67 | | 0.72 ± 0.45 | | [0.03] | 3.63 ± 0.97 | | [1.46] | 0.50 ± 0.13 | | [-0.19] | 2.58 ± 0.58 | | [0.41] |
| P04 | 0.59 ± 0.20 | | 2.29 ± 0.68 | | 0.90 ± 0.45 | | [0.31] | 3.90 ± 0.73 | | [1.61] | 0.47 ± 0.12 | | [-0.12] | 2.74 ± 0.64 | | [0.45] |
| P08 | 0.52 ± 0.17 | | 2.04 ± 0.47 | | 0.43 ± 0.41 | | [-0.09] | 3.28 ± 0.65 | | [1.24] | 0.49 ± 0.12 | | [-0.03] | 2.36 ± 0.34 | | [0.32] |
| P11 | 0.49 ± 0.12 | | 1.66 ± 0.29 | | 0.16 ± 0.16 | | [-0.33] | 2.25 ± 0.35 | | [0.59] | 0.46 ± 0.08 | | [-0.03] | 2.03 ± 0.27 | | [0.37] |
| P05 | 0.45 ± 0.20 | | 1.76 ± 0.61 | | 0.26 ± 0.20 | | [-0.19] | 2.77 ± 0.39 | | [1.01] | 0.41 ± 0.12 | | [-0.04] | 2.02 ± 0.38 | | [0.26] |
| P06 | 0.41 ± 0.11 | | 1.42 ± 0.25 | | 0.38 ± 0.29 | | [-0.03] | 3.88 ± 0.73 | | [2.46] | 0.44 ± 0.11 | | [0.03] | 2.30 ± 0.53 | | [0.88] |
| P07 | 0.41 ± 0.14 | | 1.75 ± 0.50 | | 0.26 ± 0.26 | | [-0.15] | 2.52 ± 0.74 | | [0.77] | 0.41 ± 0.11 | | [0.00] | 1.97 ± 0.45 | | [0.22] |
| P09 | 0.37 ± 0.08 | | 1.28 ± 0.19 | | 0.16 ± 0.17 | | [-0.21] | 2.35 ± 0.54 | | [1.07] | 0.39 ± 0.09 | | [0.02] | 1.84 ± 0.33 | | [0.56] |
| P10 | 0.41 ± 0.10 | | 1.46 ± 0.40 | | 0.52 ± 0.48 | | [0.11] | 3.80 ± 1.19 | | [2.34] | 0.41 ± 0.11 | | [0.00] | 2.11 ± 0.85 | | [0.65] |
| P12 | 0.34 ± 0.13 | | 1.25 ± 0.16 | | 0.11 ± 0.13 | | [-0.23] | 1.82 ± 0.35 | | [0.57] | 0.38 ± 0.09 | | [0.04] | 1.74 ± 0.30 | | [0.49] |