# Peer review of "Exploring the regolith with electrical resistivity tomography in largescale surveys: electrode spacing related issues and possibility"

_Hydrology and Earth System Sciences, 2020_

## Referee Comment (RC1) · Anonymous Referee #1 · 6 Sep 2020

P.1 - L.17: in Abstract: "larger ES" (instead of "smaller ES") P.19: I suggest that time-domain EM soundings may be very efficient for conductive bedrock recognition.

---

## Referee Comment (RC2) · Anonymous Referee #1 · 6 Sep 2020

One feels always embarassed to comment a well written and extended paper, but dealing with non-original subject and conclusions. As it the paper looks like more a master dissertation than a research paper.

While the main conclusion is that when the use of small Electrode Separation (ES) in ERT may give long acquisiton time and limited investigation depth, you can improved the ERT results when added some selected levels with ERT acquisition with larger ES. However this solution is presented on a particular 3-layer model (conductive solum- resistive subsolum- conductive bedrock), is threfore ground dependant, and the authors do not conclude with a general improvement procedure.

The paper is therefore justified by a particular case study on some hyrdogeological catchment, but the study could be more efficient (especially on a plainly tabular ground) when simply dealing with the influence of the minimum electrode spacing in vertical electrical sounding for instance, on the model resolution, especially when there is some issues with first thin layers. The particular result about the maximum optimal ES linked to first layer thickness can be quickly demonstrated without ERT and Nash-Sutcliffe stuff !

Since the paper insists on the definite influence of a priori information (the so-called "adapted vertical resolution", p.21) from the field, not only for geophysical inversion and interpretation, but also for the geophysical acquisition parameters (a criterion often forgoten for instance, in ERT acquisitions with large to very large ES) the paper seems therefore worth on-line publication.

---

## Referee Comment (RC3) · Anonymous Referee #2 · 1 Oct 2020

This manuscript presents an original study on the impact of electrode spacing (ES) on the resolution of resistivity models resulting from Electrical Resistivity Tomography (ERT) surveys. The study is illustrated by a series of synthetic data through forward modelling and by one field case study at the Weierbach catchment in Luxembourg. I must say, the technical aspects of this paper are excellent. The authors have used a state-of-the-art methodology and most of the processing steps of the ERT data, both for the forward modelling and the inversion, are relevant and well explained (sometimes too much). In fact, this is a very good technical paper. However, the research question is trivial and the proposed updated methodology is somewhat questionable. The authors even mention in the abstract that (most of) the findings are obvious (!).

Decreasing the ES, will indeed result in a greater resolution of the resistivity model, no doubts about that. Now, this paper has the merit of documenting very well the effect of different ES via a synthetic case study (Fig 3 is a great one for teaching purposes for example), and to illustrate that decreasing the ES has also an effect on the precision of the retrieved boundaries at depth (which is potentially the most interesting outcome of the paper). But then the next question is: is this paper suited for HESS? I am not too sure, since it proposes a slightly questionable updated methodology for ERT measurements done on sites with very specific characteristics in terms of homogeneously flat soil/geological structures, and for those interested to image both the thin soil layer and deeper structures. I don't see that this falls into the scope of HESS to be honest. Sure the authors have oriented the introduction on the benefits of ERT for hydrological investigations, but the rest of the paper does not really matches with HESS at this stage, even when discussing the right way of estimating precise depths of boundaries between deeper layers. I think that the methodology would be more suited for publication in a more technical journal on applied geophysics, potentially focusing primarily on the synthetic modelling. And then perhaps, building up on the published methodology, the authors could demonstrate its benefit for hydrological purposes through a real case study. But again, the technical aspects of this paper are outstanding.

Specific comments

My understanding of the proposed methodology is that it works on a site with a configuration of "conductive / resistive / conductive" three-layered structure, with a shallow layer "homogeneous in terms of resistivity and thickness." The authors propose to survey long profiles with large ES and smaller profiles with a smaller ES. And then, add some interpolated data points for shallow levels of apparent resistivity in the datasets of the long profiles. This is a bit tricky since it includes potential biases in the dataset of the long profile and poses questions in terms of spatial interpolation between smaller profiles and long profiles. Moreover, if you know that the shallow layer of your site is already homogeneous in terms of resistivity and thickness, what is the point to survey
the site? And if it is actually not homogeneous, there is a great chance that this method will virtually tell you that it is, which is more problematic. It is also not clear what is the interpolation approach used in the method. Are the authors simply extracting a mean value for different depth levels of apparent resistivity and include that in the large ES datasets as a series of virtual quadrupoles along the profile with the corresponding depth level of apparent resistivity, or are the authors spatially interpolating the apparent resistivity of several small surveys with small ES into the large ES datasets? I think a diagram explaining the methodology would be highly beneficial for the reader to understand it a bit better.

Descriptions of the synthetic results are a bit confusing, especially in section 3.1.1 which describe a lot of figures and tables (both in the manuscript and in the supplementary materials), which elongates the reading of the manuscript a bit too much. In fact, I think there are too much scenarios and results to describe for a paper. I would rather suggest the author to focus on a couple of scenarios to simplify the text.

---

## Author Comment (AC1) · 12 Nov 2020

First, we would like to thank Anonymous Referee #1 (hereafter AR1) for taking the time to read and assess our manuscript. Here below, we have reproduced all AR1's comments in normal font, followed by **our responses in bold**.

General comment

One feels always embarassed to comment a well written and extended paper, but dealing with non-original subject and conclusions. As it the paper looks like more a master dissertation than a research paper.

While the main conclusion is that when the use of small Electrode Separation (ES) in ERT may give long acquisiton time and limited investigation depth, you can improved the ERT results when added some selected levels with ERT acquisition with larger ES. However this solution is presented on a particular 3-layer model (conductive solum- resistive subsolum- conductive bedrock), is threfore ground dependant, and the authors do not conclude with a general improvement procedure.

The paper is therefore justified by a particular case study on some hydrogeological catchment, but the study could be more efficient (especially on a plainly tabular ground) when simply dealing with the influence of the minimum electrode spacing in vertical electrical sounding for instance, on the model resolution, especially when there is some issues with first thin layers. The particular result about the maximum optimal ES linked to first layer thickness can be quickly demonstrated without ERT and Nash-Sutcliffe stuff !

Since the paper insists on the definite influence of a priori information (the so-called "adapted vertical resolution", p.21) from the field, not only for geophysical inversion and interpretation, but also for the geophysical acquisition parameters (a criterion often forgotten for instance, in ERT acquisitions with large to very large ES) the paper seems therefore worth on-line publication.

**In his review, AR1 qualifies our work as well written and documented and recommends it for publication in conclusion. We would like to thank her/him for that appreciation. Nevertheless, AR1 expresses some concerns about our work which we will try to answer below.**

**AR1 expresses concerns related to a certain lack of originality of our work - both in terms of subject and subsequent conclusions. We would agree with this statement if our contribution was only about showing the influence of the electrode spacing (ES) on ERT results accuracy. However, to the best of our knowledge, no study has investigated and documented in detail from which ES threshold (as well as why and how) the accuracy of inverted ERT images is significantly affected for a given regolith structure. Eventually this aspect, which is first documented in our work, serves to introduce the problem equally dealt with in our contribution: "How to compensate for the use of an oversized ES?"**

**This issue can be particularly important for large scale ERT surveys, such as catchment scale studies, that could be really cost and time-consuming if a too small ES is to be used (e.g., due to logistical or budgetary constraints). We actually go further than showing that ERT images based on large ES can be improved by adding some selected surficial apparent resistivity levels. We believe our approach to be an innovative contribution for overcoming the bias caused by the presence of a top thin layer in a subsurface structure (e.g., a soil layer within a typical solum-to-bedrock regolith continuum).**

**In case of an ERT survey carried out with a large ES – and for which the first acquisition level does not directly give information on the resistivity of the subsurface structure's top layer – we propose to take advantage of the potential relationship between this first acquisition level and additional surficial apparent resistivity acquisition levels obtained from a reduced number of measurements with a smaller ES. We demonstrate through our work that these relationships (which are linear regressions in our study) can be strong enough if the top layer has a rather constant thickness and resistivity (such as for the solum of the Weierbach catchment). They can then be transposed to areas where solely the larger ES have been used**

and where data gaps prevail in the shallow subsurface. Our study proposes an innovative solution for improving the accuracy of ERT profiles based on large ES.

As discussed in our manuscript, and as stated by AR1, we agree that the proposed upgrading approach is not a general improvement procedure at this stage (we focused on a specific conductive-resistive-conductive 3-layer structure) and it depends on site specificities (i.e., the top layer has to be rather homogeneous, compared to the underlying layers). Note that in our manuscript we actually invite the reader to confirm/infirm the proposed methodology for the reverse case "resistive solum / conductive subsolum / resistive bedrock" in other study sites. We believe that our methodology is not "Weierbach catchment – specific". As discussed in our contribution, the regolith of the Weierbach catchment is representative of the slate regolith which covers a large part of the Rhenish Massif. Hence, we anticipate that the proposed protocol could be used across many regions of this large central European geological area (extending from Luxembourg, through Belgium, France and Germany) and might thus be of interest for the hydrological sciences community working in this region. Moreover, as written in our manuscript, we further expect that our novel approach may also be transferable beyond this area to other regions/catchments with similar characteristics, like forested catchments with similar bedrock geology.

AR1 suggests also that our study would have been more efficient if a Vertical Electrical Sounding (VES) logic and a 1D forward/inverse modelling approach had been used for the synthetic case study. We disagree with AR1 on this point as the use of a 1D modelling approach is less informative. It is worth recalling that we have opted for the use of a 1D synthetic model structure, but that the subsequent forward modelling and inversion processes have then been done in 2D in order to evaluate not only the accuracy, but also the precision of ERT inversion results. This would not have been possible using a 1D inversion scheme. We therefore think that our choice to use a 2D modelling approach for the synthetic case in order to deal with a 2D ERT specific issue is fully justified.

Misprints and comments

P.1 - L.17: in Abstract: "larger ES" (instead of "smaller ES")

We thank AR1 for pointing out this mistake. If our work is accepted for publication in HESS, this will be obviously corrected in the revised version of the manuscript.

P.19: I suggest that time domain EM soundings may be very efficient for conductive bedrock recognition.

We agree with AR1 that ground-based TDEM soundings could be suitable for detecting the bedrock of the Weierbach catchment (or in a similar context). However, although this technique might potentially provide results as precise as those derived from ERT, to the best of our knowledge there is no fast-moving device allowing quick measurements. We thus believe that TDEM is not more time efficient than ERT to explore accurately the regolith over large areas for a comparable horizontal sampling resolution. This statement is also strengthened by Figure 5 in Binley et al. (2015), who compare the horizontal and vertical survey scales typically achievable in 1 day by a two person field crew for ERT, FDEM, and TDEM.

**References**

Binley, A., Hubbard, S. S., Huisman, J. A., Revil, A., Robinson, D. A., Singha, K., and Slater, L. D.: The emergence of hydrogeophysics for improved understanding of subsurface processes over multiple scales, Water Resources Research, 51, 3837-3866, 2015.

---

## Author Comment (AC3) · 12 Nov 2020

First, we would like to thank Anonymous Referee #2 (hereafter AR2) for taking the time to read and assess our manuscript. Here below, we have listed all AR2's comments in normal font, followed by **our responses in bold**.

General comment

This manuscript presents an original study on the impact of electrode spacing (ES) on the resolution of resistivity models resulting from Electrical Resistivity Tomography (ERT) surveys. The study is illustrated by a series of synthetic data through forward modelling and by one field case study at the Weierbach catchment in Luxembourg. I must say, the technical aspects of this paper are excellent. The authors have used a state-of-the-art methodology and most of the processing steps of the ERT data, both for the forward modelling and the inversion, are relevant and well explained (some-times too much). In fact, this is a very good technical paper. However, the research question is trivial and the proposed updated methodology is somewhat questionable. The authors even mention in the abstract that (most of) the findings are obvious (!). Decreasing the ES will indeed result in a greater resolution of the resistivity model, no doubts about that. Now, this paper has the merit of documenting very well the effect of different ES via a synthetic case study (Fig 3 is a great one for teaching purposes for example), and to illustrate that decreasing the ES has also an effect on the precision of the retrieved boundaries at depth (which is potentially the most interesting outcome of the paper). But then the next question is: is this paper suited for HESS? I am not too sure, since it proposes a slightly questionable updated methodology for ERT measurements done on sites with very specific characteristics in terms of homogeneously flat soil/geological structures, and for those interested to image both the thin soil layer and deeper structures. I don't see that this falls into the scope of HESS to be honest. Sure the authors have oriented the introduction on the benefits of ERT for hydrological investigations, but the rest of the paper does not really matches with HESS at this stage, even when discussing the right way of estimating precise depths of boundaries between deeper layers. I think that the methodology would be more suited for publication in a more technical journal on applied geophysics, potentially focusing primarily on the synthetic modelling. And then perhaps, building up on the published methodology, the authors could demonstrate its benefit for hydrological purposes through a real case study. But again, the technical aspects of this paper are outstanding.

**AR2 qualifies our work as technically outstanding and very well documented. We would like to thank her/him for that assessment. However, we also identify in AR2's general comment some concerns about our study that we want to address below.**

**AR2 states that our manuscript might be a good technical paper dedicated to geophysicists, but that it eventually does not perfectly fit to HESS. It is true that the technical aspect dealing with Electrical Resistivity Tomography (ERT) limitation/improvement is central to our study, but in our opinion the guiding idea behind our work – which is to provide new insights into subsurface mechanisms and aid in the parameterization of subsurface flow and transport models – is inextricable and just as important. For that reason, we believe that our research work must be considered as a hydrogeophysical study rather than dealing with a technical geophysical problem alone. As documented in the introduction of our manuscript, several authors have recently pointed out the subsurface as being the greatest knowledge gap in the understanding of hydrological processes, with a greater investment into "seeing" the subsurface needed to provide the Earth System Modelling community with critical guidance on how to parametrize model subsurface structure depths and properties. This general framework has strengthened our belief that the topic of our work is well-suited to HESS. We are thus convinced that our research work provides results and findings that are of interest to and can be used by the wide research community targeted by HESS (see Discussion section 4.3 From the Weierbach catchment perspective and beyond).**

**AR2 also suggests that our work might be somewhat trivial. Indeed, he/she states that the impact of ES on inverted ERT images accuracy is already a well-known issue. While we fully agree of course with AR2 on this statement, we nonetheless thought that it would be worth mentioning this obvious fact, while targeting a non-geophysicist readership that might be less familiar with this technique. But our work does**

not focus on this general issue, but rather addresses a specific and less trivial aspect. As indicated in our introduction, we were wondering whether deep structures are well defined if the shallow structure is not well sampled. Our work was specifically concentrating on the influence of the ES on ERT results accuracy when a top thin layer is present in a subsurface structure (e.g., the soil layer within a typical soil-to-substratum continuum). To the best of our knowledge, no study has documented so far in detail from which ES threshold (as well as why and how) the accuracy of inverted ERT images is significantly affected for a typical regolith structure. Note that such a layered sequence of soil–saprock/saprolite–bedrock mirrors the subsurface of many natural contexts. We found out from our study that the thickness of the most superficial layer (i.e., the soil in our case) must be considered when choosing the ES – even when solely aiming for the characterization of deeper layers. We are convinced that this key result is of importance for the hydrological sciences community in search of accurate characterizations of regolith geometry and properties. Our approach ultimately untaps the potentially biased results (in terms of both resistivity distribution and interface delineation) that ERT would provide if this condition is not satisfied. Note that this outcome is also recognised by AR2.

Finally, AR2 is questioning the significance of our novel upgrading approach as it addresses sites with specific characteristics. It is correct that we assessed this new methodology on one tabular three-layer structure. However, as explained in our manuscript, this structure was chosen because representing a typical regolith sequence of soil–saprock/saprolite–bedrock that mirrors the subsurface of many natural areas. The upgrading procedure was experienced on a specific conductive-resistive-conductive 3-layer sequence in order to mirror the Weierbach catchment case study setting. This approach may also work in other contexts, such as for example the reverse case, i.e. "resistive solum / conductive subsolum / resistive bedrock". Note that we especially invite the potentially interested readership for assessing the proposed methodology in this latter situation. Nevertheless, we agree with AR2 that the proposed upgrading approach is not a general improvement procedure as it is applicable at sites with a rather homogeneous top layer. However, we also want to stress that the proposed methodology is not "Weierbach catchment – specific". As discussed in our manuscript, the regolith of the Weierbach catchment is representative of the slate regolith which covers a large part of the Rhenish Massif. Hence, we anticipate that the proposed protocol could be used in manifold sites of this large central European geological area (extending from Luxembourg, through Belgium, France and Germany). Therefore, the proposed approach shall eventually be of interest for the hydrological sciences community working in this region. Moreover, as written in our manuscript, we further expect that our novel approach may also be transferable to other regions/catchments with similar characteristics, like forested catchments with similar bedrock geology.

Specific comments

My understanding of the proposed methodology is that it works on a site with a configuration of "conductive / resistive / conductive" three-layered structure, with a shallow layer "homogeneous in terms of resistivity and thickness." The authors propose to survey long profiles with large ES and smaller profiles with a smaller ES. And then, add some interpolated data points for shallow levels of apparent resistivity in the datasets of the long profiles. This is a bit tricky since it includes potential biases in the dataset of the long profile and poses questions in terms of spatial interpolation between smaller profiles and long profiles. Moreover, if you know that the shallow layer of your site is already homogeneous in terms of resistivity and thickness, what is the point to survey the site? And if it is actually not homogeneous, there is a great chance that this method will virtually tell you that it is, which is more problematic. It is also not clear what is the interpolation approach used in the method. Are the authors simply extracting a mean value for different depth levels of apparent resistivity and include that in the large ES datasets as a series of virtual quadrupoles along the profile with the corresponding depth level of apparent resistivity, or are the authors spatially interpolating the apparent resistivity of several small surveys with small ES into the large ES datasets? I think a diagram explaining the methodology would be highly beneficial for the reader to understand it a bit better.

From the specific comments above, we understand that on the basis of the current version of our contribution, any potential reader of our manuscript might miss a key step proper to the proposed upgrading procedure. The proposed methodology is indeed based on the incorporation of "virtual quadrupoles" defining several shallower levels of apparent resistivity. However, these virtual levels neither are constant mean values calculated for the entire study area nor result from a spatial interpolation between available shallow apparent measurements measured with a smaller ES. As explained in section 2.3 (step 3 of the protocol), ERT profiles using an oversized ES might be upgraded with surficial levels of apparent resistivity interpolated from a) calibrated relationships which were defined using a reduced number of shallow apparent resistivity measurements done with a smaller ES and b) the values of its own first acquisition level of apparent resistivity as input of the calibrated relationships. It is in this aspect that the innovation of the method lies. In the Weierbach catchment for instance, the four linear regressions resulting from the plot scale ERT profiles dataset done with an ES of 0.5 m (Figure 9) are intended to be used all over the catchment area. Thus, we plan to upgrade the ERT profiles measured with an ES of 2 m and covering all the catchment (white lines drawn in Figure 1) using as input the values of their own first acquisition levels of apparent resistivity (i.e. quadrupoles with external electrodes separation of 6 m). Note that we believe that the proposed upgrading procedure is well described in section 2.3, but we will try to clarify the text and/or introduce an additional diagram to better explain the procedure if requested.

AR2 mentions also that our novel procedure could introduce potential biases in the upgraded dataset. We are not exactly sure which bias AR2 is referring to. However, note that we already highlighted limitations of the proposed method, as well as potential improvements to face it, in the discussion section 4.2. For instance, we pointed out some specific local areas in the Weierbach catchment where the method would most probably lead to erroneous results by inducing false inverted surficial resistivity layers (e.g. riparian zone, where solum and subsolum have been eroded). It goes without saying that the upgrading protocol will not be applied to these locations. But we want to remind that our study clearly demonstrated that the application of the proposed upgrading method to the catchment-wide ERT survey dataset relying on an ES of 2 m will lead to an overall improvement of the inverted results accuracy. As shown in our work, if we do not apply this procedure, the lack of shallow apparent resistivity related to the oversized ES of 2 m data will for sure induce a much more important general bias.

AR2 finally asks why we want to survey a site if we already know that the shallow layer is rather homogeneous. We assume that AR2 refers to the solum of the Weierbach. As indicated in our manuscript, the solum itself is not the primary target of our catchment scale ERT survey. The goal is to inform on the spatial variability of the regolith as a whole: weathering state, potential relation with hydraulic properties, depth to bedrock, spatial organisation/connectivity, new insights on the substratum further deep.

Descriptions of the synthetic results are a bit confusing, especially in section 3.1.1 which describe a lot of figures and tables (both in the manuscript and in the supplementary materials), which elongates the reading of the manuscript a bit too much. Infact, I think there are too much scenarios and results to describe for a paper. I would rather suggest the author to focus on a couple of scenarios to simplify the text.

To shorten sections 3.1.1 and 3.1.2, we are willing to describe in the manuscript only the Wenner-Schlumberger array results if requested. Results obtained for the dipole-dipole array, which are similar to Wenner-Schlumberger ones, might only be presented in the supplementary materials. However, as proposed by AR2, we disagree to reduce the number of synthetic scenarios studied. It is indeed the diversity of these scenarios with varying resistivity and thickness contrasts that leads to a comprehensive understanding of the impact of the lack of shallow apparent resistivity data induced by the use of an oversized ES on inverted ERT image accuracy.

---

## Author Response (AR2)

Dear editor, dear referee,

Thank you for this second opportunity to revise our manuscript. Please find below Anonymous Referee #2's comments (in normal font), followed by our responses (in bold) and the location of the changes we made in the manuscript (underlined).

Kind regards,
On behalf of all co-authors
Laurent Gourdol
* * *
**Anonymous Referee #2**

I would like to thank the authors for this revised version of the manuscript, and more specifically for the detailed answers to the initial comments. The authors have addressed a series of concerns on the originality of the study, and I must admit that their justification as to why this work is worth publishing in HESS is totally reasonable. I have no major objections anymore for this study to be considered for publication in HESS (after clarifying some remaining points of concerns). However, is worth mentioning that even though I'm satisfied with the author's rebuttal, the revised version of the manuscript remains in essence rather similar to the first version, and was not substantially modified, as one would have expected after a "major revision".

**We would like first to thank Anonymous Referee #2's (hereafter AR2) for the additional time spent on reviewing our work and for providing thoughtful suggestions of improvement.**

The proposed novel procedure for ERT measurements, which is the main focus of the manuscript, and described in Sect. 2.3 remains too poorly explained in this revised version of the manuscript, and I'm afraid is still subject to misunderstanding. The new figure 4 illustrating the novel approach is clearly a really good addition to help capturing the main concept, but, apologies for this, I still struggle to understand the method. The authors state in their answer to the editor that in the original version of the manuscript, the reader might miss a key step of the proposed procedure. The new version of the manuscript does not seem to bring much more explanation on this key step, as no improvement of the manuscript has been included in Sect. 2.3 of the methodology to support the new Figure 4. The missing key point concerns what data are used in the linear regressions. It is indeed not clear what data are plotted against each other on the scatter plots (Fig. 6 and Fig. 10). I know realise that this might come from a missing information in the first step detailed in Sect. 2.3, which states that "from the set of apparent resistivity data measured with an ES of 2 m, we extract the first acquisition level of apparent resistivity data (for the smallest possible external electrodes separation, i.e. 6 m). For this acquisition level, we extract – from the set of apparent resistivity data relying on an ES of 0.5 m – four subsets of apparent resistivity data for smaller external electrodes separations of 1.5, 2.5, 3.5 and 4.5 m, respectively." There is no information as to what these subsets are made of. Is it the apparent resistivity of the measurement array which has the pseudo-x position closest to that of the array with the larger ES? Is it an average of the apparent resistivities for all the arrays of a specific smaller ES comprised within the larger ES? As things stand, the reader can only guess what this subset is made of, and I would suggest the authors to explain in much more detail how it is extracted. It wouldn't arm anyway to have

a slightly more detailed Sect 2.3 of the manuscript. This is the claimed main novelty of the study, so I would really strengthen the explanation of the method. In the current version of the manuscript, there is a rather shortly explained method, with lengthy examples, but if the method is not clear in the first place, the examples are not really helpful. Clarifying how the subsets are extracted in the legends of Fig. 6 and Fig 10 would also be welcomed.

**We agree with AR2's assessment and admit that the explanation of our upgrading methodology in section 2.3 is not clear/detailed enough and might be consequently misleading for the reader. According to AR2's suggestions, we revised section 2.3 as well as captions of Figure 6 and 10. We also modified the caption of Figure S14 in the supplement.**

A practical aspect of the proposed method could also be included in the discussions. Indeed, have the authors investigated the appropriate number of ERT profiles with smaller ES needed to improve significantly the accuracy of the ERT measurements with larger ES? It could be interesting to assess if using a smaller number of ERT profiles than the 12 field datasets gives similar results. Since the area seems to have a rather constant solum thickness, could only 1 ERT profile with smaller ES provides a robust linear regression? This aspect might be critical to assess the practical feasibility of the method, and is worth including in the discussion.

**We thank AR2 for this good point. Unfortunately, we did not investigate precisely the impact of the number of ERT profiles using an ES of 0.5 m on the accuracy of the upgraded ERT profiles relying on the ES of 2 m for the Weierbach catchment. The 12 plot scale ERT profiles were selected based on our *a priori* knowledge, in order to have a set of spatially well distributed profiles over the catchment area, and covering the range of prevailing local geomorphological characteristics. However, as asked specifically by AR2, we can assert that using a single ERT profile with an ES of 0.5 m would be highly uncertain in the Weierbach catchment, despite the fact that this study area exhibits a rather homogeneous solum in terms of thickness and resistivity. The ability of the calibrated linear regressions relying on one single profile to deliver accurate prediction for the full data set is indeed highly variable from one profile to another as shown in the figure on the next page. Although some profiles lead to predictions comparable to those obtained from the complete dataset, others provide poor results. We nevertheless agree with AR2 that the number of accurate ERT profiles with the small ES (together with their location) is of key relevance for the upgrading procedure to succeed. On the one hand, as highlighted by AR2, if this one is too large, the applicability of the upgrading procedure might be cumbersome. On the other hand (and most importantly in our opinion), if it is too small, the upgrading of the ERT profiles relying on the large ES might lead to less reliable and inaccurate inverted results, or even biased results. Practically, for both cases, it is about evaluating the efficiency of the dataset used for calibration to deliver robust estimates of the surficial levels of apparent resistivity regarding the dataset size. New elements have been added to the penultimate paragraph of the discussion subsection 4.2 in order to deal with this aspect, as well as a new citation has been added to the reference list.**

**Finally, it is worth noting that we indicated in our revised manuscript the contribution and competing interests of the authors after the conclusion. A distinct data availability section has also been created (the data availability was described in the Acknowledgments of the previous version of our manuscript).**

[Figure]

Scatter plots relating the apparent resistivity data corresponding to the first pseudo-depth acquisition level for an ES of 2 m (external electrodes spacing of 6 m) versus the shallower first four surficial apparent resistivity levels for an ES of 0.5 m with external electrodes separations of 1.5 (a), 2.5 (b), 3.5 (c) and 4.5 m (d) for the 12 plot scale ERT profiles measured in the Weierbach catchment. Each of the linear regressions presented in this figure (colored lines) was calibrated using one single ERT profile (one colour represents one specific profile). Their accuracy indicated in brackets (root mean square error) was however computed by considering all the data of the 12 ERT profiles.